# MRI signatures of cortical microstructure in human development align with oligodendrocyte cell-type expression

Sila Genc [1,2,3] ✉, Gareth Ball [2,4], Maxime Chamberland [1,5], Erika P. Raven [1,6,7], Chantal M. W. Tax [1,8], Isobel Ward [1,9], Joseph Y. M. Yang [2,3,4,10], Marco Palombo [1,11] & Derek K. Jones [1]

Neuroanatomical changes to the cortex during adolescence have been well documented using MRI, revealing ongoing cortical thinning and volume loss. Recent advances in MRI hardware and biophysical models of tissue informed by diffusion MRI data hold promise for identifying the cellular changes driving these morphological observations. Using ultra-strong gradient MRI, this study quantifies cortical neurite and soma microstructure in typically developing youth. Across domain-specific networks, cortical neurite signal fraction, attributed to neuronal and glial processes, increases with age. The apparent soma radius, attributed to the apparent radius of glial and neuronal cell bodies, decreases with age. Analyses of two independent post-mortem datasets reveal that genes increasing in expression through adolescence are significantly enriched in cortical oligodendrocytes and Layer 5–6 neurons. In our study, we show spatial and temporal alignment of oligodendrocyte cell-type gene expression with neurite and soma microstructural changes, suggesting that ongoing cortical myelination processes drive adolescent cortical development.

Over the last two decades, magnetic resonance imaging (MRI) has provided invaluable insights into the developing brain, revealing ongoing cortical thinning and cortical volume loss throughout adolescence[1,2]. However, the underlying cellular processes driving these changes are less understood. Cortical cytoarchitecture can be broadly categorised into neurites (e.g., axons, dendrites, and glial processes) and soma (e.g., neuronal, and glial cell bodies). Traditionally, synaptic pruning has been considered the primary driver of developmental changes in cortical morphology[3]. Recent evidence, however, suggests

that myelin encroachment into the grey/white matter boundary may also contribute to changes in MR contrast typically used for volumetrics, such as $T_1$[4]. Developmental patterns of cortical myelination have been elucidated using magnetisation transfer (MT) imaging[5], and indirectly using T1w/T2w ratio[6]. Despite these advances, how microstructural changes – specifically neurite and soma properties – contribute to these distinct morphological changes remains unclear.

Diffusion-weighted MRI (dMRI) is the main non-invasive MRI technique capable of probing the tissue microstructure, orders of

[1]Cardiff University Brain Research Imaging Centre (CUBRIC), School of Psychology, Cardiff University, Cardiff, UK. [2]Developmental Imaging, Clinical Sciences, Murdoch Children's Research Institute, Parkville, VIC, Australia. [3]Neuroscience Advanced Clinical Imaging Service (NACIS), Department of Neurosurgery, The Royal Children's Hospital, Parkville, VIC, Australia. [4]Department of Paediatrics, University of Melbourne, Parkville, VIC, Australia. [5]Department of Mathematics and Computer Science, Eindhoven University of Technology, Eindhoven, The Netherlands. [6]Department of Radiology, NYU Grossman School of Medicine, New York, NY, USA. [7]Institute for Translational Neuroscience, NYU Grossman School of Medicine, New York, NY, USA. [8]Image Sciences Institute, University Medical Center Utrecht, Utrecht, The Netherlands. [9]Population Health Sciences, Bristol Medical School, University of Bristol, Bristol, UK. [10]Neuroscience Research, Clinical Sciences, Murdoch Children's Research Institute, Parkville, VIC, Australia. [11]School of Computer Science and Informatics, Cardiff University, Cardiff, UK. ✉e-mail: sila.genc@mcri.edu.au

magnitude smaller than the typical millimetre image resolution of structural MRI[7]. This microstructural imaging method is highly sensitive to the magnitude and direction of water diffusing within brain tissue. By employing biophysical models, it is possible to infer microscopic properties of different tissues, such as neurite signal fraction in the brain's white matter[8,9]. In comparison with white matter, grey matter cytoarchitecture, broadly categorised into neurites (e.g., elongated structures such as axons, dendrites and glial processes) and soma (e.g., spherical structures such as neuronal and glial cell-bodies) is more locally complex, requiring extensions of the standard models of microstructure developed for studying the white matter. Recent hardware[10,11] and biophysical modelling[12–14] developments have enabled diffusion-weighted microstructural quantification of soma and neurite components in the cortex in vivo. The Soma and Neurite Density Imaging (SANDI; Palombo, Ianus[13]), is robust, reliable[15], clinically feasible for sufficiently short diffusion times[16] and has been validated in ex vivo data[17].

SANDI is a biophysical tissue model that estimates the diffusion-weighted signal contribution from three distinct compartments: intra-neurite, intra-soma, and extracellular space. For each imaging voxel, a signal fraction will be estimated for each of the three compartments, such that they sum to 1. In the cortical grey matter, there is a higher proportion of soma (neuronal and glial cell bodies) to neurites, leading to a higher soma signal fraction. These signal fractions vary around tissue boundaries, with higher extracellular signal fraction around the cortical surface due to partial voluming with CSF. Overall, these compartment-specific signal fractions are relative, and comparing these trends over age are potentially meaningful to deduce the compartments that are contributing most to age-related changes in cortical development.

Here, we examine cortical microstructural development in a sample of children and adolescents using ultra-strong gradient dMRI to identify specific changes in neurite and soma properties with age. To identify potential cellular substrates, we analyse developmental patterns of neurite and soma microstructure alongside contemporaneous trajectories of cortical cell-type specific gene

expression measured in the developing cortex using data from two independent, post-mortem databases. We reveal key developmental patterns in cortical neurite and soma architecture, highlighting the contribution of active and ongoing cortical myelination processes to the macroscale changes observed in the cortex during adolescence.

## Results

We apply a framework for cortical microstructure and cell-type specific gene expression analysis (Fig. 1) to evaluate the cellular properties underpinning human cortical microstructural development.

### Cortical microstructure and morphology in domain-specific networks

First, we studied the repeatability of cortical microstructural estimates from the SANDI model in a sample of 6 healthy adults scanned over 5 sessions. Intra-class coefficients (ICCs) for neurite signal fraction ($f_{neurite}$; mean ICC = 0.97), soma signal fraction ($f_{soma}$; mean ICC = 0.98) and extracellular signal fraction ($f_{extracellular}$; mean ICC = 0.98) were very high (Fig. 2c) across seven domain-specific networks. Apparent soma radius ($R_{soma}$, in μm) showed lower repeatability on average (mean ICC = 0.94) with lower mean repeatability driven by the limbic network.

We then studied age-related patterns of cortical microstructure and morphology in a sample of 88 typically developing children and adolescents aged 8–19 years (Table S2). Cortical $f_{neurite}$ and intracellular volume fraction ($v_{ic}$; derived from the NODDI model, Zhang, Schneider[8]) increased with age across all cortical networks (mean $R^2_{fneurite}$ = 0.53, all networks $p < 3.3e\text{-}11$; mean $R^2_{vic}$ = 0.46, all networks $p < 1.6e\text{-}9$) (Fig. 2d, Fig S1). Orientation dispersion index (ODI; derived from the NODDI model, Zhang, Schneider[8]) also increased with age across all studied networks (mean $R^2_{odi}$ = 0.42, all networks $p < 1.9e\text{-}5$). In contrast, we observed decreasing $R_{soma}$ with age across all networks (mean $R^2_{Rsoma}$ = 0.48, all networks $p < 4.4e\text{-}10$) and $f_{soma}$ decreased with age in the dorsal attention ($R^2_{fsoma}$ = 0.12), limbic ($R^2_{fsoma}$ = 0.09) and somatomotor ($R^2_{fsoma}$ = 0.23), networks (all $p < 0.002$). $f_{extracellular}$ decreased in the default mode ($R^2_{fe}$ = 0.12), limbic ($R^2_{fe}$ = 0.21) and

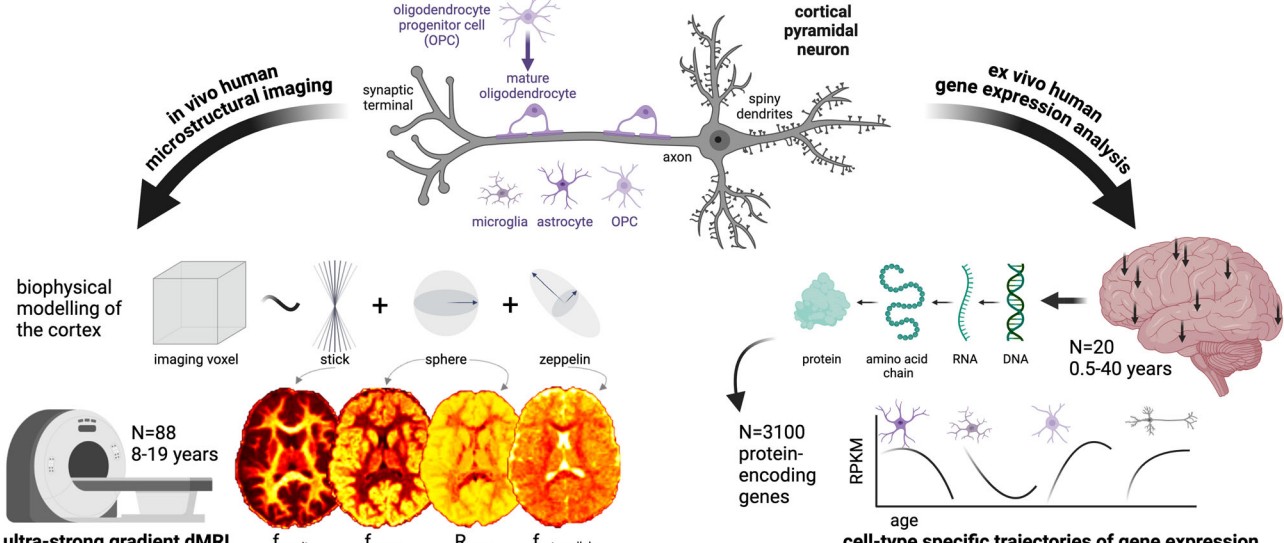

**Fig. 1 | Framework for cortical microstructure and gene expression analysis.** This study employs a biophysical model of cortical neurite and soma microstructure using ultra-strong gradient dMRI[10] data collected from 88 children and adolescents aged 8–19 years. Representative maps of neurite signal fraction ($f_{neurite}$), soma signal fraction ($f_{soma}$), apparent soma radius ($R_{soma}$, μm) and extracellular signal fraction ($f_{extracellular}$) are shown for one 8-year-old female participant. We also analyse two human gene expression datasets[20,21] to estimate cell-type

specific and spatial (where arrows on brain render indicate a subset of regions sampled) gene expression profiles and examine their concordance with developmental patterns of cortical microstructure. Created in BioRender. Genc, S. (2025) https://BioRender.com/q33l208. Abbreviations: dMRI: diffusion magnetic resonance imaging; $f_{extracellular}$: extracellular signal fraction; $f_{neurite}$: neurite signal fraction; $f_{soma}$: soma signal fraction; RPKM: rates per kilobase of transcript per million mapped; $R_{soma}$: apparent soma radius, in μm.

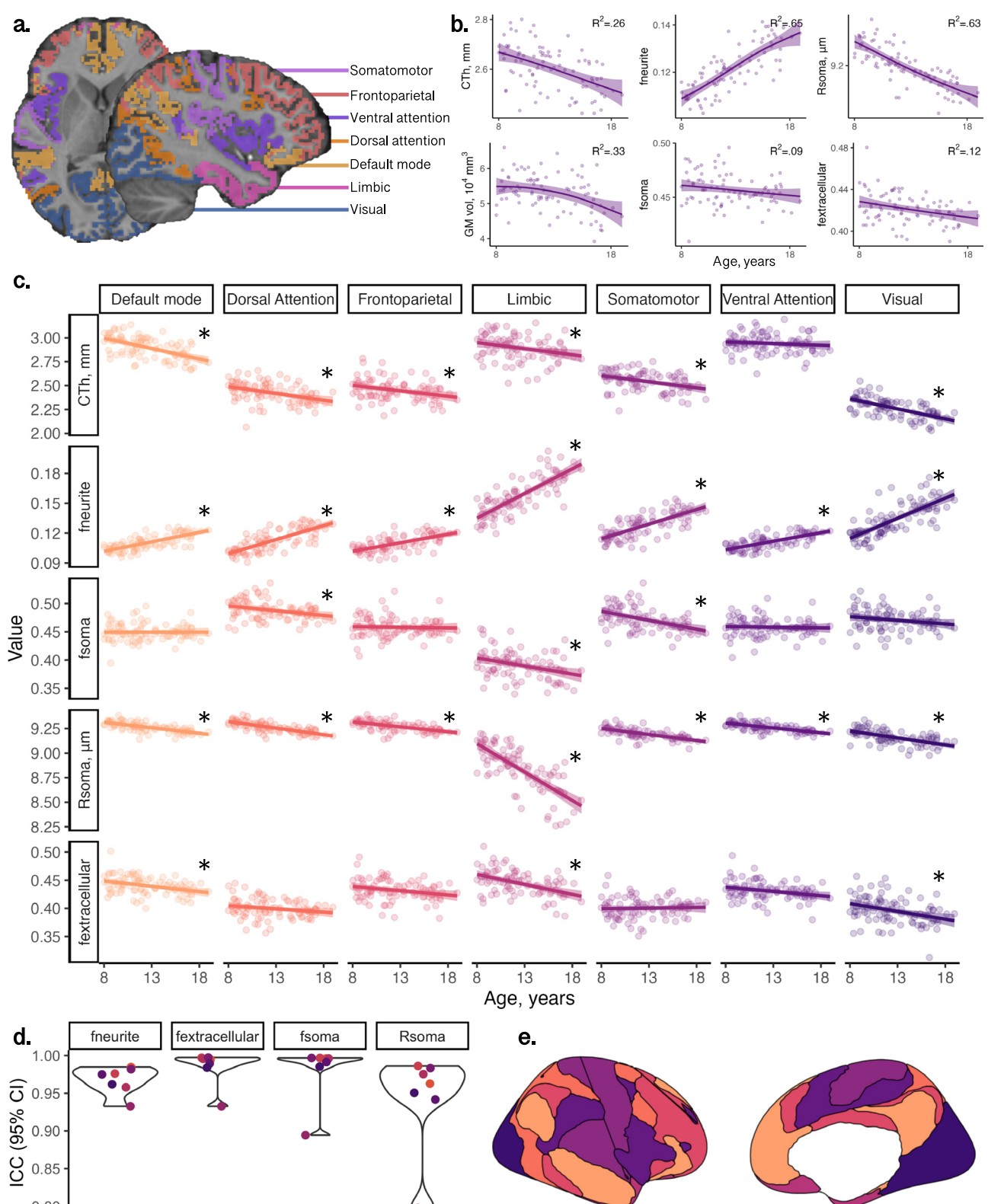

**Fig. 2 | Developmental patterns of MRI-derived cortical morphology and microstructure. a** regions in atlas used to derive domain-specific networks[81] overlaid on a representative participant; **b** developmental patterns of cortical morphology and microstructure averaged across the cortical ribbon; **c** network-wide patterns of microstructure and morphology, indicating age-related increases in neurite fraction and reductions in cortical thickness, apparent soma radius, soma fraction and extracellular fraction; **d** demonstration of high repeatability of SANDI measures in six adults scanned over 5 time-points within two weeks; **e** spatial representation of networks. Significant age relationships determined with a linear regression ($p < 0.005$) are annotated (*) and exact $p$ values are reported in Table S2. Curves in (**b**, **c**) represented as mean trajectory with 95% confidence interval bounds. Abbreviations: CTh cortical thickness, in mm, $f_{extracellular}$ extracellular signal fraction, $f_{neurite}$ neurite signal fraction, $f_{soma}$ soma signal fraction, GM grey matter, ICC intra-class coefficient, $R_{soma}$ apparent soma radius, in μm. Colour coding in (**a**, **e**) corresponds to regions in (**c**, **d**). Source data for (**d**) are provided as a Source Data file.

visual ($R^2_{fe} = 0.09$) networks (all $p < 0.004$). DTI metrics revealed decreasing FA with age across all networks apart from the limbic network (mean $R^2 = 0.27$, $p < 2.1e\text{-}5$), decreasing MD in the limbic network, $\beta = -0.42$, [−0.62, −0.22], $p = 8.8e\text{-}5$, and increasing MD in the somatomotor network, $\beta = 0.34$, [0.13, 0.55], $p = 0.002$.

Consistent with established developmental patterns, cortical thickness and grey matter volume decreased with age (Fig. 2b). The strength of these associations varied across brain networks (see Fig S1 and Table S2). Specifically, cortical thickness exhibited age-related decline in the default mode, $\beta = -0.59$ [−0.77, −0.41], dorsal attention, $\beta = -0.40$ [−0.61, −0.19], somatomotor, $\beta = -0.40$ [−0.60, −0.19], and visual, $\beta = -0.61$ [−0.78, −0.43], networks (all $p < 0.001$). Similarly, grey matter volume decreased with age in the default mode, $\beta = -0.37$ [−0.55, −0.20], dorsal attention, $\beta = -0.34$ [−0.54, −0.15], and visual $\beta = -0.29$ [−0.47, −0.11], networks (all $p < 0.002$). Cortical surface area did not show significant age-related differences. The magnitude and direction of age effects across all microstructural and morphological measures are shown in Fig. S1.

Sex differences in brain structure been well reported, with pubertal onset playing a critical role in initiating developmental changes to morphology[18] and microstructure[19]. We found that grey matter volume and surface area were higher in males than females ($p < .005$) across all brain networks (Fig. S2), following known patterns of larger brain volume in males. We observed sex differences in only two microstructural measures, $R_{soma}$ and fractional anisotropy (FA; derived from the diffusion tensor at b = 1200 s/mm$^2$), in the visual network (Fig. S2, S3). Females had higher $R_{soma}$, $\beta = -0.57$ [−0.91, −0.24], $p = 0.001$, and lower FA, $\beta = 0.55$, [0.18, 0.92], $p = 0.004$, compared to males. We observed a pubertal stage by sex interaction on $f_{soma}$, where males had lower soma signal fraction in early puberty, $\beta = 0.73$ [0.28, 1.18], $p = 0.002$, which stabilised in late puberty. Males had lower $f_{extracellular}$ throughout puberty $\beta = -0.74$ [−1.18, −0.31], $p = 0.001$. To further demonstrate developmental differences in the visual network we built an age prediction model (see supplementary section 8.3.2 and Fig. S10) which showed that $R_{soma}$ provided the most accurate age-prediction (cross-validated $R^2 = 0.58$). Feature importance revealed top-ranking regions represent cortical endpoints of developmentally sensitive tracts, identified through tractography, such as the posterior corpus callosum, cingulum, and inferior longitudinal fasciculus (Fig. S10d).

### Contemporaneous gene expression trajectories

Using $n = 214$ post-mortem tissue samples from the dorsolateral prefrontal cortex (DLFPC; BrainCloud; Colantuoni, Lipska[20]), we identified $n = 2057$ genes with differential expression over the lifespan (0.5–72 years; $p_{FDR} < 0.05$). We validated this selection in an independent RNA-seq dataset (PsychENCODE; Li, Santpere[21]; $n = 20$), identifying $n = 467$ (22.7%) genes with significant age-associations in both datasets (age-genes; Supp Info).

We identified sets of differentially expressed genes across 7 cortical cell-types (see Methods). Mean trajectories of gene expression across ages 0 and 30 years, averaged within each cell-type, are shown as standardised curves in Fig. 3 for PsychENCODE (Fig. 3a) and Brain-Cloud (Fig. 3b) datasets. Developmental profiles from the DLPFC were visualised, to allow for clearer comparisons of cell-type specific trends over age between cohorts. Non-normalised gene expression curves for PsychENCODE are presented in Fig. S4 to aid in interpreting relative differences in gene expression magnitudes. Among genes expressed in excitatory neuronal populations and oligodendrocytes, mean expression levels increased with age. In contrast, genes expressed in inhibitory neurons showed no age-related variation. Genes expressed in endothelial cells, astrocytes, microglia and OPCs, exhibited a decrease in mean gene expression with age. Overall, microglial gene expression (mean log$_2$RPKM = 1.96) was lower compared to astrocytes (mean log$_2$RPKM = 3.70), oligodendrocytes (mean log$_2$RPKM = 3.11), OPCs

(mean log$_2$RPKM = 3.01), excitatory neurons (mean log$_2$RPKM = 4.15) and inhibitory neurons (mean log$_2$RPKM = 2.94). To validate our bulk-tissue findings in an independent dataset, we took advantage of a recent single-cell RNA atlas of pre- and postnatal brain development[22]. Using these data, we identified a set of cell-specific genes with an onset of expression in childhood (>4 years) followed by a rapid increase through adolescence and into adulthood ($n = 534$ genes). Most of these genes were expressed by oligodendrocytes ($n = 349$; Fig. S12), confirming our findings from bulk-tissue data.

We confirmed the enrichment of cell-types identified in the age-related genes identified using bulk-tissue data using an independent cell-type specific expression analysis (CSEA). Significant enrichment of age-genes ($n = 467$) was observed in cortical oligodendrocytes, oligodendrocyte progenitors, and Layer 5–6 neurons (Fig. S5). These genes were prominently expressed across developmental stages in childhood adolescence, and young adulthood (Fig. S6, Table S3, Fig. 3c). The number (Fig. 3d) and proportion (Fig. 3e) of age-related genes expressed by oligodendrocytes increased significantly in adolescence and young adulthood (Fig. 3d, e). These included genes associated with CNS (re)myelination, RCAN2[23], GRIA3[24], and the differentiation of OPCs and oligodendrocytes, PLEHA1/TAPP1[25]; AATK/AATYK[26].

For each cell-type, we quantified the spatiotemporal patterns of gene expression using PsychENCODE data by identifying the peak growth of expression in cell-specific genes. Oligodendrocyte gene expression peaked earliest in primary motor (M1), primary visual (V1) cortices, and latest in the medial frontal (MFC) cortex (Fig S8). A notable pattern emerged in which the peak expression of oligodendrocyte genes coincided with a shift in oligodendrocyte-to-astrocyte specific expression ratio. This shift, indicating a relative increase in oligodendrocyte over astrocyte cell-type gene expression, occurred around 20 years of age in M1 and V1, and after age 25 in DLPFC, ITC and MFC (Fig. 3g, h). This sequence aligns with the known earlier myelination timing in sensorimotor cortices followed by prolonged myelination in the pre-frontal cortex into the third decade of life[5,6,27].

### Concordant profiles of microstructure and gene expression indicate developmental cortical myelination

To elucidate the cell-specific basis of our imaging findings, we examined neurite and soma microstructural measures in the same four frontal regions sampled in the PsychENCODE data (MFC, IFC, DLPFC, VLPFC; see Fig. 4a, b) using a fine-grained parcellation of the frontal lobe. Microstructural MRI revealed regional increases in $f_{neurite}$ and decreases in $R_{soma}$ (Fig. 4c). This pattern corresponded with increased regional oligodendrocyte cell-type gene expression profiles in the same regions over the same age period (Fig. 4d, e). The spatial distribution of oligodendrocyte cell-type expression was aligned with regional differences in peak growth of the neurite fraction (Fig. S7). Thus, the dMRI-derived neurite signal fraction likely reflects spatiotemporal patterns of cortical myelination, matching the peak expression of oligodendrocyte-genes.

To further evaluate the concordance between in vivo MRI and ex vivo gene expression patterns, we performed numerical simulations using realistic cell counts to explain age-related patterns of the apparent soma radius. Assuming that the observed age-related slope of gene expression was proportional to the number of cells of each cell-type within an MRI voxel, we modelled cell-type composition changes based on the actual expected distribution of cell body radii within a voxel based on realistic cell counts and sizes. Our results revealed close correspondence between simulated and in vivo modelling results of $R_{soma}$ (Fig. S9), showing a 1% age-related decrease in both simulated and dMRI-derived data across 8–19 years.

## Discussion

We combined in vivo ultra-strong gradient dMRI with independent ex vivo gene expression analyses to map tissue microstructural

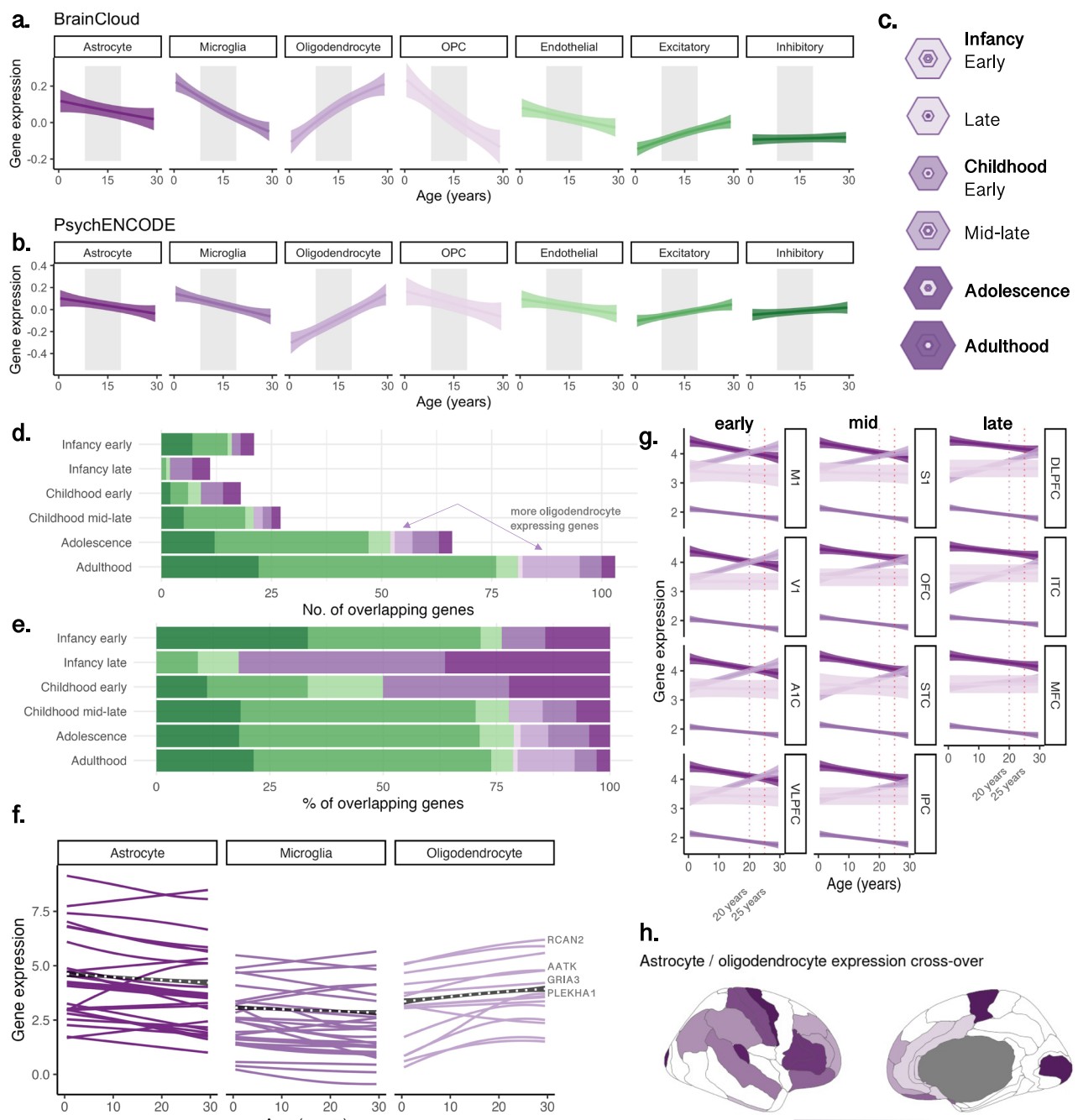

**Fig. 3 | Developmental trajectories of cell-type specific gene expression.** Data shown for samples aged 0-30 years from: (**a**) BrainCloud (Z-score), and (**b**) PsychENCODE (expressed in log₂-reads-per-kilobase of transcript per million (log₂RPKM)) datasets, demeaned to account for overall higher expression in some cell-types. Age effects were modelled in all postnatal samples to maximise sample size. Grey shaded areas highlight the age range of the microstructural imaging cohort (8–19 years) for visual comparison of developmental profiles. **c** SEA results[89] showing significant enrichment of age-related genes through adolescence and adulthood, where hexagon size scales with enrichment (overlap) of age-related genes in genes expressed by each cell type determined using the Fisher's exact test[91], and darker rings indicate significant associations at $p_{FDR} < .001$ with inner rings indicating high cell specificity. False discovery rate (FDC) was controlled using Benjamini-Hochberg multiple testing correction for the number of cell types and regions assayed[92]. Age-related genes overlapping postnatal developmental stages are shown as (**d**) total number of genes, and (**e**) proportion of genes, indicating an increase in neuronal, glial and oligodendrocyte-specific genes. **f** Trajectories of glial genes overlapping the SEA and our age-genes. **g** Regional shifts in the glial cell-type expression ratio (log₂RPKM) across development, with the astrocyte-to-oligodendrocyte expression ratio crossing earliest at age 20 years in primary motor and visual cortices. **h** Timing of this cross-over, with darker values indicating regions with an earlier crossing point. Note that white coloured regions are not represented in the data set. Curves in (**a**, **b**, **g**) represented as mean trajectory with 95% confidence interval bounds. Abbreviations: A1C Primary auditory cortex, DLPFC Dorsolateral pre-frontal cortex, IPC Inferior parietal cortex, ITC Inferior temporal cortex, M1 Primary motor cortex, MFC Medial frontal cortex, OFC Orbito-frontal cortex, OPC oligodendrocyte precursor cell, RPKM rates per kilobase of transcript per million mapped, S1 Primary somatosensory cortex, STC Superior temporal cortex, V1 Primary visual cortex, VLPFC Ventrolateral pre-frontal cortex. Colour coding in (**a**) corresponds to cell-types in (**b**, **d**–**f**). Source data are provided as a Source Data file.

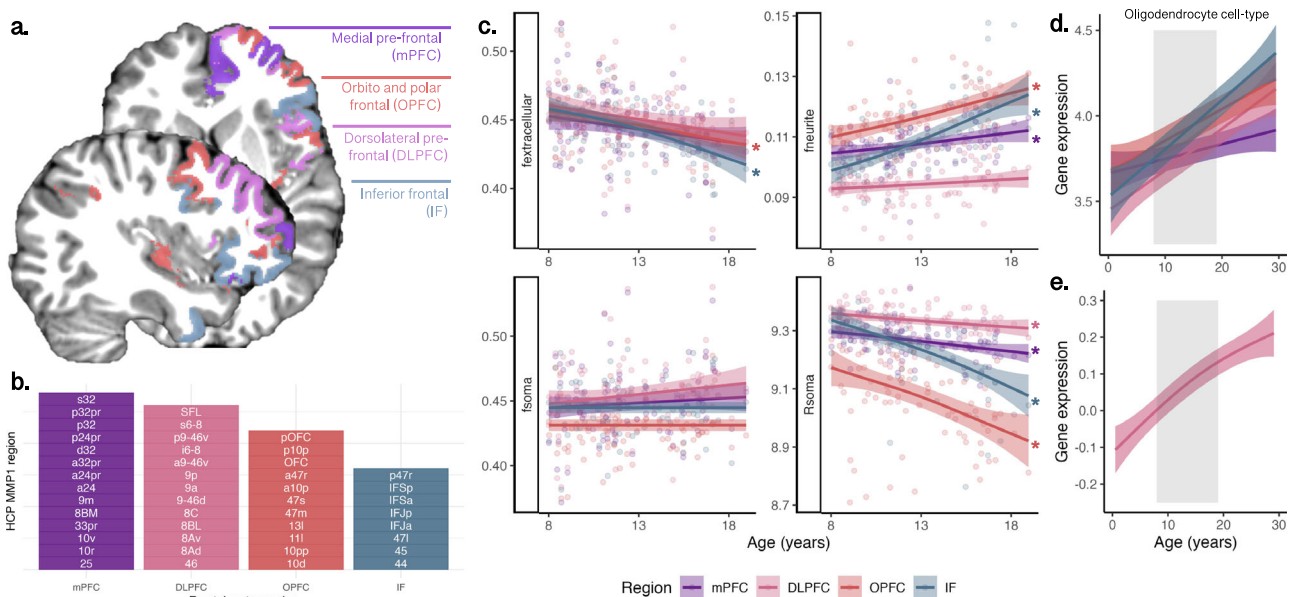

**Fig. 4 | Regional variation of microstructure and gene expression in the frontal cortex. a** Structural MRI-based segmentation of four frontal regions: medial pre-frontal cortex (mPFC); dorsolateral prefrontal cortex (DLPFC); orbito and polar frontal cortex (OPFC), and inferior frontal cortex (IF); **b** Sub-regions from the HCP-MMP1 atlas[82], which comprised the regions in (**a**); **c** Age-related patterns of microstructural measures with significant age relationships determined with a linear regression: $f_{extracellular}$ (mPFC: $p =$ ; OPFC: $p = 0.000194$; DLPFC: $p =$ ; IF: $p = 3.2e-7$); $f_{neurite}$ (mPFC: $p = 0.00169$; OPFC: $p = 1.09e-6$; IF: $p = 4.8e-11$); $R_{soma}$

(mPFC: $p = 0.000173$; DLPFC: $p = 0.00828$; OPFC: $p = 5.81e-06$; IF: $p = 1.66e-09$); Oligodendrocyte cell-type gene expression in (**d**) PsychENCODE data sampled in the same 4 frontal cortical regions as (**a**, **e**) BrainCloud data sampled in the DLPFC. Curves represented as mean trajectory with 95% confidence interval bounds. Abbreviations: $f_{extracellular}$ extracellular signal fraction, $f_{neurite}$ neurite signal fraction, $f_{soma}$ soma signal fraction, $R_{soma}$ apparent soma radius, in μm. Colour coding in (**a**) corresponds to regions in (**b**–**e**). Source data for (**d**, **e**) are provided as a Source Data file.

architecture during human development. We now discuss each of the key findings and their implications, before summarising the strengths and limitations of our study.

### Neurite signal fraction increases from childhood to adolescence

The neurite signal fraction, $f_{neurite}$, attributed to elongated cortical structures (e.g., axons, processes), increased with age across the whole cortex, but peaked earliest in the visual and somatomotor networks, mirroring previous findings[28]. Intracortical myelination continues over adolescence[4,6,29–31], following a stereotyped sensorimotor-to-association (S-A) axis of development[32]. Although dMRI is relatively insensitive to water within the myelin sheath itself, due to its short $T_2$[33], the observed increase in $f_{neurite}$ may nevertheless reflect intra-cortical myelination. In dMRI, myelin thickening can decrease the extracellular signal fraction, due to less physical space in the extracellular matrix[33,34]. Age-related decreases in $f_{extracellular}$ were confined to the visual net-work and orbito-frontal and inferior frontal cortices. Ex vivo macaque data support developmental increases in glial process length and complexity[35], and an increase in the number of myelinated axons and dendrites[36], which limits water exchange and leads to a greater signal contribution from inside the neurite[14,37]. Comprehensive evaluation of the myelin content is warranted to confirm the contributions of intracortical myelination to developmental changes in cortical morphology[38].

### Oligodendrocyte-specific gene expression increases from childhood to adolescence

Supporting our in vivo MRI findings, oligodendrocyte-specific gene expression increased with age (Fig. 3a, b), aligning with previous observations in independent data[5]. Age-related genes were also enri-ched in cortical neurons (layers 5 and 6) and OPCs (Fig. S5). The con-cordance between the human bulk-tissue gene expression analysis (Fig. S5), human single-cell RNA analysis (Fig. S12), and the CSEA ana-lysis based on mouse transcriptomic profiling (Fig. S5) indicates

conservation of myelination processes via cortical oligodendrocytes. Oligodendrocyte cell turnover in the frontal cortex is dynamic, espe-cially in adulthood, and 10 times higher in the cortex than in the white matter[39]. OPCs can generate myelinating oligodendrocytes in adult-hood, even in fully myelinated regions[40,41]. Importantly, oligoden-drocyte function is not restricted to myelination, rather, they also perform many critical neuronal support functions beyond myelination[42]. Together our microstructural MRI and gene-expression findings converge towards increased cortical myelination through adolescence.

### Apparent soma radius decreases from childhood to adolescence

The dMRI-derived apparent soma radius, $R_{soma}$, decreased cortex-wide from childhood to adolescence. Neuronal soma are much larger than glial soma, measuring ~16 μm in diameter in layers 5–6 of the adult human prefrontal cortex, whereas glial soma range in diameter from 1 to 11 μm[43]. Our gene expression analysis suggests specific changes in the cellular composition of the cortex with age: decreasing expression levels for astrocyte, microglia and endothelial cell-types, and (much larger) increasing expression levels for oligodendrocyte cell-types. Glial composition in the neocortex is mostly comprised of oligoden-drocytes (~75%), followed by astrocytes (~20%) and a smaller pre-valence of microglia (~5%)[44]. Assuming gene expression levels are proportional to cell number/density, our observations suggest a decrease in large-soma cells (e.g., endothelial), outweighed by a larger increase in small-soma cells (e.g., oligodendrocytes).

The estimated $R_{soma}$ is dependent on the higher order moments of the soma radii distribution (i.e., skewdness and tailedness) within an MRI voxel[37]. Our own simulations of $R_{soma}$ based on known cell com-position in the human brain[45] revealed a decrease in apparent soma radii with age matching our in vivo imaging observations (i.e., a 1% decrease). This would in turn lead to a reduction in the measured dMRI signal coming from water molecules fully restricted in soma, aligning with our in vivo observations of decreasing $f_{soma}$ with age in the limbic,

somatomotor, and dorsal attention networks. It is plausible that an increase in oligodendrocyte[46], not astrocyte or microglial[35], composition could concomitantly result in a smaller average soma radii and lower soma signal fraction in the cortex through adolescence to early adulthood.

Females have larger apparent soma radii than males, and $f_{soma}$ and $f_{extracellular}$ varies with pubertal stage in the visual network (Fig. S10a). Pubertal hormones can stimulate apoptosis (seen in female rat visual cortex; Nunez, Sodhi[47]), which could explain the lower $f_{soma}$ as puberty progresses in females. Selective neuronal cell death with unchanged glial cell number can also occur during puberty in the medial pre-frontal cortex[48,49], however we did not observe any sex or pubertal differences in microstructure of the frontal cortex.

## Spatiotemporal patterns of gene expression
Peak oligodendrocyte cell-type gene expression progressed along the S-A axis, with earliest peaks in M1 and V1, and latest in MFC (Fig. S8), mirroring spatial patterns of peak $f_{neurite}$ (Fig. S7). This also coincided with a relative age-related decrease in astrocyte cell-type gene expression (Fig. 4g) consistent with early-life maturation of astrocytes[50,51]. The S-A developmental axis describes a maturation process from lower-order, primary sensory and motor (unimodal) cortices to higher-order transmodal association cortices, which support complex neurocognitive, and socioemotional functions[27,52]. Prolonged maturation of the pre-frontal cortex has been reported with lower myelin content in fronto-polar cortex compared with visual or somatomotor regions from childhood to adulthood[53] indicating later myelination timing. Within the frontal cortex, age-related patterns of microstructural neurite signal fraction and soma radius were prolonged in the MFC and DLPFC (Fig. 4c–e). This reflects the value of estimating in vivo neurite signal fraction as these developmental hierarchies have been reproduced across various modalities[27,54–58], particularly when considering the regions reaching peak maturation earliest and latest. Overall, our combined imaging genetic analyses supports the evidence of an orderly and hierarchical progression of intracortical myelination.

## Potential applications
The findings of our work has implications for the study of cortical thinning. A recent study showed that cortical thinning during development is associated with genes expressed predominantly in astrocytes, microglia, excitatory and inhibitory neurons[59]. We observed faster cortical thinning of default-mode and visual networks, consistent with previous studies[59,60]. Apparent thinning may be a result of the macrostructural shift in the boundary between grey matter and white matter, in this scenario due to myelin encroachment into the cortex[4,61]. The microstructural composition of the grey matter itself may be better studied by the biophysical models used here.

Cortical morphology and myelination abnormalities are linked to various neuropsychiatric disorders[62] including schizophrenia[63,64] which is characterised by deficiencies in myelination and oligodendrocyte production[65,66]. One potential future application is to quantify cortical microstructure in such clinical cohorts, especially with adaptations towards clinically feasible acquisition protocols using current state-of-the-art clinical grade 3 T systems[16,67,68], and with the recent advent of commercial systems with ultra-strong gradients (e.g., Siemens 3 T Cima.X; GE 3 T MAGNUS). Further strengthening this potential application, schizophrenia patients exhibit downregulation of myelination-related genes[69] and post-mortem studies have shown reduced oligodendrocyte density in layer 5 of dorsolateral prefrontal cortex compared to healthy controls[70]. Additionally, young children with autism show age-related deficits in cortical T1w/T2w ratios[71]. Future studies exploring these neuroimaging measures may provide valuable insights into cortical based abnormalities.

## Strengths and limitations
Several methodological advancements have advanced the understanding of underlying compositional changes to cortical microstructure across development in our study. Our repeatability results show that SANDI-derived biophysical signal fractions are highly stable (mean ICC = 0.97) in a young adult population, and these values are highly concordant with recently reported cortex-wide measurements in a subset of younger adults[72]. Using in vivo microstructural imaging with ultra-strong gradients ($G_{max}$ = 300 mT/m; Jones, Alexander[10]), we achieved sensitivity to micrometre-level imaging contrast by maximising SNR and minimising the effect of water exchange[73]. Although we used a specialised system, recent advancements have enabled these measurements on more accessible, lower-gradient strength MRI systems (e.g., $G_{max}$ ≥ 80mT/m; Schiavi, Palombo[16]). Combined with two ex vivo gene expression data sets sampled from the human brain, we provide compelling evidence in favour of a framework for monitoring intra-cortical cellular composition in vivo. Further work should evaluate in vivo imaging acquisition techniques and models that account for water exchange, which can influence biophysical modelling of grey matter compartments.

Our observation of oligodendrocyte-specific gene expression increasing towards adulthood indicates the value of imaging a broader age range of young adults to fully assess trajectories of in vivo microstructural properties. It is also important to recognise that gene expression patterns do not necessarily correlate with cellular density. Histopathological confirmation is needed to verify cell size and density with biophysical signal fractions, as well as their relevancy to functional gene expression patterns.

Overall, our study provides in vivo evidence of distinct developmental differences in neurite and soma architecture, aligning with cell-type specific gene expression patterns observed in ex vivo human data. This provides a window into the role of intracortical myelination through adolescence, and how it shapes the developmental patterns of cortical microstructure in vivo.

## Methods
### Ethics
Imaging data acquisition was performed as part of the Cardiff University Brain Research Imaging Centre (CUBRIC) Kids study approved by the School of Psychology ethics committee at Cardiff University. All procedures were completed in accordance with the Declaration of Helsinki.

### Imaging set
**Participant characteristics.** We included a sample of 88 typically developing children aged 8–19 years (42 males) recruited as part of the Cardiff University Brain Research Imaging Centre (CUBRIC) Kids study. Participants and their parents/guardians were recruited via public outreach events. Written informed consent was obtained from the primary caregiver of each child participating in the study, and adolescents aged 16–19 years also provided written consent. Verbal assent was obtained for participants younger than 16 years. Children were excluded from the study if they had non-removable metal implants or reported history of a major head injury or epilepsy. Participants and their families were reimbursed with a £20 gift voucher for their time and participation.

We administered a survey to parents of all participants, and to children aged 11–19 years. The Strengths and Difficulties Questionnaire (SDQ) was used to assess emotional/behavioural difficulties (Goodman, 1997). The Pubertal Development Scale[74] was administered which asks questions specific to female and male sex characteristics and used to determine pubertal stage (PDSS; Shirtcliff, Dahl[75]). Additionally, we measured each child's height and weight to calculate their Body-Mass index (BMI) (kg/m²). Table 1 summarises the cohort characteristics.

**Table 1 | Characteristics of in vivo imaging cohort**

| Measure | Summary statistics | | | Age relationship[d] | |
|---|---|---|---|---|---|
| | Mean | SD | Range | R² | p-value |
| Age, years[a] | 12.56 | 2.94 | 8.0–19.0 | | |
| Pubertal stage (PDSS)[a] | 2.89 | 1.50 | 1–5 | 0.72 | 2e-16 |
| SDQ, total score[a] | 6.45 | 3.90 | 0–19 | 0.01 | 0.60 |
| Body mass index, kg/m²[b] | 19.29 | 3.25 | 13.7–29.2 | 0.25 | 1.6e-6 |
| FSIQ[c] | 108 | 12.6 | 86–145 | | |

[a]Full sample: N = 88 (42 males, 46 females).
[b]Subsample: N = 79 (40 males, 39 females).
[c]Subsample: N = 48 (23 males, 25 females).
[d]Age relationships determined by linear regression.

All children and adolescents underwent in-person training to prepare them for the MRI procedure using a dedicated mock MRI scanner. This protocol was 15–30 min. long, and designed to familiarise them to the scanner environment, to minimise head motion during the scan.

**Acquisition and processing.** Discovery data: Participants aged 8–19 years (N = 88, mean age = 12.6 years, 46 female) underwent MRI on a 3 T Siemens Connectom system with ultra-strong (300 mT/m) gradients. Structural $T_1$-weighted (voxel-size = $1 \times 1 \times 1$ mm³; TE/TR = 2/2300 ms) and multi-shell dMRI (TE/TR = 59/3000 ms; voxel-size = $2 \times 2 \times 2$ mm³; $\Delta$ = 23.3 ms, $\delta$ = 7 ms, b-values = 0 (14 vols), 500, 1200 (30 dirs), 2400, 4000, 6000 (60 dirs) s/mm²) data were acquired. Data were acquired in an anterior–posterior (AP) phase-encoding direction, with one additional PA volume. The total acquisition time (across four acquisition blocks) was 16 min 14 s.

Repeatability data: Six healthy adults aged 24–30 years (3 female) were scanned five times in the span of two weeks[76] on the same Siemens Connectom system. Multi-shell dMRI data were collected as above, with an additional 20 diffusion directions acquired at b = 200 s/mm². One participant had missing T1 data in one MRI session so the data from that single session was excluded due to the inability to perform cortical parcellation.

Pre-processing of dMRI data followed steps interfacing tools such as FSL (v6.0.5)[77], MRtrix3 (v3)[78], and ANTS (v2.1.0)[79] as reported previously[80]. Briefly, this included denoising, and correction for drift, motion, eddy, and susceptibility-induced distortions, Gibbs ringing artefact, bias field, and gradient non-uniformities. For each subject, the soma and neurite density imaging (SANDI) compartment model was fitted[13] to dMRI data using the SANDI Matlab Toolbox v1.0, publicly available at https://github.com/palombom/SANDI-Matlab-Toolbox-v1.0, to compute whole brain maps of neurite, soma and extracellular signal fraction ($f_{neurite}$, $f_{soma}$, $f_{extracellular} = 1 - f_{neurite} - f_{soma}$); the apparent soma radius ($R_{soma}$, in μm); and the extracellular and intra-neurite axial diffusivities ($D_e$ and $D_{in}$, respectively, in μm²/ms) (Fig. 1, Fig S11). To put our results in context with previous studies, the neurite orientation dispersion and density imaging (NODDI) model[8] was fitted to all b-values using the NODDI Matlab toolbox, publicly available at http://mig.cs.ucl.ac.uk/index.php?n=Tutorial.NODDImatlab, to estimate the intra-cellular volume fraction ($v_{ic}$) and orientation dispersion (OD) and diffusion tensor imaging (DTI) metrics were estimated using the b = 1200 s/mm² shell (Fractional anisotropy (FA); mean diffusivity (MD, in s/mm²).

$T_1$-weighted data were processed using FreeSurfer (v6.0; http://surfer.nmr.mgh.harvard.edu) and post-processed to obtain network-level (N = 7 ROIs; Yeo, Krienen[81]) and fine-grained cortical parcellations (N = 360, HCP-MMP1[82];). The network-level atlas derivation is detailed in Genc, Schiavi[83]. Briefly, we co-registered the Yeo functional atlas in MNI space to each individual subject's space, to obtain seven

functionally relevant cortical canonical networks (visual, somato-motor, dorsal attention, ventral attention, limbic, frontoparietal, default mode network). We chose a functional atlas due to the limitations of structural atlases in capturing fine-grained microstructural variations, enabling better insights into developmental patterns of neural activity and connectivity[84]. Follow-up analyses using fine-grained HCP-MMP1 parcellations in visual and frontal cortices were performed based on a priori hypotheses of earlier maturation of visual[4] and later maturation of frontal[35] cortices, as well as for comparison with gene expression data sampled from multiple regions in the frontal cortex. Morphological measures including cortical thickness (CTh, mm), surface area (SA, mm²), and grey matter volume (GMvol, mm³) were computed at the whole brain, and parcel level. The analysis framework is detailed in Fig. 1 and networks studied are depicted in Fig. 2a.

**Cortical gene expression set.** Pre-processed, batch-corrected and normalised microarray and bulk RNA-seq data from postmortem human tissue samples were obtained from the BrainCloud[20] (n = 214; aged 6mo – 78.2 y; 144 male; postmortem interval [PMI] = 29.96 [15.28]; RNA integrity [RIN] = 8.14 [0.83]) and PsychENCODE (n = 20; 6mo–40y; 10 male; PMI = 17.85 [6.75]; RIN = 8.45 [0.79]) projects, respectively[21]. The cortical regions sampled are summarised in Table S1, alongside the approximate concordant Yeo7 parcel. Tissue processing is detailed elsewhere[21,85]. Gene expression for PsychENCODE was measured as rates per kilobase of transcript per million mapped (RPKM). Gene expression for Braincloud was preprocessed and normalised following data cleaning and regressing out technical variability (see https://www.ncbi.nlm.nih.gov/geo/query/acc.cgi?acc=GSE30272).

Genes were initially filtered to include only protein-coding genes expressed in cortical cell types (n = 3100, Ball, Seidlitz[85]). Using a database of single-cell RNA-seq studies, we identified genes differentially expressed across major cortical cell types (excitatory and inhibitory neurons, oligodendrocytes, oligodendrocyte precursor cells [OPCs], microglia, astrocytes, and endothelial cells[60]).

**Statistical analyses**
**In vivo imaging.** For the repeatability analysis, the intra-class correlation coefficient (ICC; two-way random effects, absolute agreement) was computed for assessment of test-re-test repeatability for SANDI and DTI metrics using the 'psych' package in R.

We used linear regression to test for main effects of age and sex, puberty, and sex by puberty interactions. To identify the most parsimonious model and to avoid over-fitting, we used the Akaike Information Criterion (AIC)[86], selecting the model with the lowest AIC. Individual general linear models were used to determine age-related differences in cortical thickness and microstructural measures in all seven Yeo networks. Evidence for an association was deemed statistically significant when $p < 0.005$[87]. Results from linear models are presented as the normalised coefficient of variation (β) and the corresponding 95% confidence interval [lower bound, upper bound]. We also report the adjusted correlation coefficient of the full model ($R^2$).

To identify important regions that contribute to age-related differences in all the studied microstructural measures, we performed age-prediction using a random forest (RF) regressor (5-fold cross-validation) for age prediction with PyCaret (www.pycaret.org). We chose a RF model due their ability to model nonlinear relationships, reduced risk of overfitting, and interpretability of feature importance. Specifically, the depth at which a feature appears as a decision node in a tree provides insight into its relative significance for predicting the target variable (i.e., age). This allows us to assess the relative contribution of features (i.e., average signal fraction in each HCP-MMP1 parcel) to age prediction. For each microstructural measure, we

randomly split the data into training and validation sets using an 80–20 ratio (total N = 88: 70 training; 18 testing). Then, we performed feature scaling to ensure that all input variables (for each HCP-MMP1 ROI) were on a similar scale prior to model fitting. The performance of the model was evaluated on the validation dataset. Finally, the features with the largest weight coefficients were extracted to identify specific cortical regions where variance in cortical microstructure was associated with age-related changes.

**Gene expression profiles.** To identify genes differentially expressed over age ($p_{FDR} < 0.05$), we modelled age-related changes in normalised expression in all available postnatal tissue samples using nonlinear generalised additive models with thin plate splines (k = 5)[88] in R.

For BrainCloud data, the relationship between normalised gene expression and age was modelled with a nonlinear general additive model (GAM) using a penalised thin-plate spline with a maximum 5 knots:

$$gam(expression \sim 1 + s(age, k = 5, bs = 'tp') \tag{1}$$

Note that the available BrainCloud data are already preprocessed to remove variance due to batch and sample effects (see https://www.ncbi.nlm.nih.gov/geo/query/acc.cgi?acc=GSE30272).

For PsychENCODE data, we repeated the above models, now with a measure of RNA integrity (RIN) as a confounder, and gene expression defined as $log_2(RPKM)$. First, we included region as an additional factor to account for spatial variation across the cortex and included donor ID as a random effect to account for repeated samples from the same specimen.

$$gam(expression \sim 1 + s(age, k = 5, bs = 'tp', by = region, id = 1) + RIN \\ + sex + region + s(sample, bs = 're'), data = data)) \tag{2}$$

Then, we analysed data only in the DLPFC, for comparison with the BrainCloud geneset.

$$gam(expression(DLPFC) \sim 1 + s(age, k = 5, bs = 'tp') + RIN + sex \\ + s(sample, bs = 're'), data = data)) \tag{3}$$

We calculated measures of goodness of fit using Akaike Information Criterion (AIC) and Bayesian Information Criterion (BIC) for all gene models.

Using a set of independent single-cell RNA studies of the human cortex (see Ball, Seidlitz et al. (2020) for details), we identified genes exhibiting differential expression across various cortical cells-types, including excitatory neurons, inhibitory neurons, oligodendrocytes, microglia, astrocytes, and endothelial cells. We then compiled gene lists for each cell-type, comprising genes that are both differentially expressed by that cell-type, and uniquely expressed by that cell-type. Mean trajectories across all cortical regions sampled were computed for each cell-type.

After identifying age-related genes, we entered our list to an independent cell-type specific expression analysis (CSEA; Xu, Wells[89]) to elucidate: (1) if genes were enriched for specific cell-types, and (2) in which developmental period was gene expression highest.

**Simulations**

We performed numerical simulations using realistic cell counts to explain the observed trends in $R_{soma}$ derived from in vivo dMRI data. We modelled the variability in cell body sizes within an MRI voxel by generating distributions of radii for microglia, astrocytes, oligodendrocytes, neurons, and endothelial cells. For each cell-type, we assumed the observed age-related slope of gene expression was proportional to the number of cells within an MRI voxel. Based on realistic cell counts outlined in Keller, Erö[45], we set the number of cells

in mm³ as follows: $N_{micro} = 6500$; $N_{astro} = 15,700$; $N_{oligo} = 12,500$; $N_{neuro} = 92,000$; $N_{endo} = Nneuro*.35$[90]. For each cell type, we generated random samples of radii based on the specified cell counts assuming a Gaussian distribution with cell-type specific baseline mean and standard deviation: microglia = $2.0 \pm 0.5\,\mu m$; astrocytes and oligodendrocytes = $5.5 \pm 1.5\,\mu m$; neurons = $8.0 \pm 2.0\,\mu m$ for neurons and $9.0 \pm 0.5\,\mu m$ for endothelial. The resulting radii were concatenated to form a comprehensive distribution and the MR apparent soma radius $R_{soma}$ estimated as $\left(\frac{R_s^5}{R^3}\right)^{1/2}$ as per Olesen, Østergaard[37].

### Reporting summary
Further information on research design is available in the Nature Portfolio Reporting Summary linked to this article.

## Data availability
Original datasets are accessible through the original publications, including the MICRA[76] neuroimaging repeatability dataset (osf.io/z3mkn/), PsychENCODE Human mRNA-seq processed data (Gene expression in RPKM: development.psychencode.org) and BrainCloud (https://www.ncbi.nlm.nih.gov/geo/query/acc.cgi?acc=GSE30272). Source data to generate figures from openly available data are provided with this paper. Due to the inclusion of minors (under 18 participants) in the MRI portion of our study, the availability of derived or identifiable data from our participant cohort is restricted due to privacy concerns. Derived data supporting the findings of the imaging analyses are available by contacting the corresponding author in writing via email (Dr Sila Genc: gencs@cardiff.ac.uk), allowing four weeks for access requests to be granted. Source data are provided with this paper.

## Code availability
Code to perform the gene expression analysis are provided as R scripts.

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

## Acknowledgements

The authors would like to thank the families that participated in this study for their generous contributions. We would also like to thank Umesh Rudrapatna and John Evans for their assistance with acquisition proto-cols and Greg Parker and Simona Schiavi for assistance with image processing. Fig. 1 was created with BioRender.com with elements from Macrovector via freepik. The imaging data were acquired at the UK National Facility for In Vivo MR Imaging of Human Tissue Microstructure funded by the EPSRC grant EP/M029778/1 and The Wolfson Foundation [D.K.J.]. The Royal Children's Hospital Foundation (RCHF 2022-1402) supported S.G. and J.Y.M.Y. The National Health and Medical Research Council Investigator Grant (1194497) supported G.B. NICHD (F32HD103313) and NIA (1R21AG083539-01A1) of NIH supported E.P.R. A Veni grant (17331) from the Dutch Research Council (NWO) and the Wellcome Trust (215944/Z/19/Z) supported C.M.W.T. UKRI Future Lea-ders Fellowship (MR/T020296/2) supported M.P. This research was funded in whole, or in part, by a Wellcome Trust Investigator Award (096646/Z/11/Z) awarded to D.K.J.) and a Wellcome Trust Strategic Award (104943/Z/14/Z) awarded to D.K.J.). For the purpose of open access, the author has applied a CC BY public copyright licence.

## Author contributions

S.G. and D.K.J. conceptualised the problem. S.G., E.R., and I.W. acquired the developmental MRI data. S.G., M.C., and M.P. analyzed the MRI data. S.G., G.B., and M.C. performed statistical analyses. D.K.J. supervised and raised funding for this project. S.G. wrote the original draft of the manuscript. S.G., G.B., M.C., E.R., C.M.W.T., I.W., J.Y.M.Y., M.P., and D.K.J. reviewed and edited the manuscript.

## Competing interests

The authors declare no competing interests.
