## [Transparent Peer Review file · Nature Communications]

MRI signatures of cortical microstructure in human development align with oligodendrocyte cell-type expression

Corresponding Author: Dr Sila Genc

Version 0:

Reviewer comments:

Reviewer #1

(Remarks to the Author)

Using ultra-strong gradient MRI to obtain high-resolution, in vivo estimates of cortical neurite and soma microstructure in typically developing children and adolescents, the authors examined the development of cortical microstructure and its underlying cellular mechanisms. The authors report that the neurite signal fraction, attributed to neuronal and glial processes increased with age whereas the soma radius decreased with age across brain networks. To complement the findings, the authors also examined developmental patterns of cortical gene expression in two independent post-mortem databases. Results showed increased expression of genes expressed in oligodendrocytes, and excitatory neurons, and a decrease in expression of genes expressed in astrocyte, microglia and endothelial cell-types in adolescence and young adulthood. The authors report that the developmental patterns in cortical neurite and soma highlight the contribution of active myelination processes in cortical maturation during adolescence.

The authors are attempting to answer a very important question in neuroscientific literature regarding the maturation of cortical microstructure during a critical period of human development. Further, they are investigating the development using a multiple model approach, combining ultra-strong gradient imaging and as well as transcriptomics, which is very exciting, as combining different methods can allow us to dig deeper into the underlying cellular and molecular mechanisms making impactful discoveries in the field of developmental neuroscience.

That said, I have a few comments, queries, and clarification on the methods and analysis that the authors used in the current study. I specifically found the analysis lacking a logical flow and confusing. I think the authors are analyzing several morphological and microstructural metrics which are important, however, there is a lack of clear reasoning of why the authors selected these metrics, or brain areas, and opens several concerns which I list below:

Methodological concerns.

1. Concerns with Figure 1: I really like the schematic in Fig. 1, however, it needs color bars telling the reader what the brain maps indicate, what do the warmer colors indicate in each map? What is the range of numbers per metric and what do they indicate exactly? The authors are using novel methods and metrics to examine cortical development, and it would be beneficial for the reader to know the range of fractions for $f_{neurite}$, f_{soma} , and $f_{extracellular}$ and radius range for R_{soma} within a voxel. Additionally, a paragraph on how these fractions /measures are obtained will be helpful and what information each measure provides will be also helpful. I would also show a comparison here between the 8-year-old sample child and an adult from one of 6 adults to help the reader understand where and what the development looks like on each of these maps.
2. In the first analysis, the authors examined the repeatability of cortical microstructural estimates from the SANDI model in a sample of 6 healthy adults scanned over 5 sessions. This is a useful analysis, but it is unclear why the authors choose to report an intra-class coefficient (ICCs) per neurite signal fraction ($f_{neurite}$), soma signal fraction (f_{soma}) and extracellular signal fraction ($f_{extracellular}$) metric and how this is calculated and why the authors chose the 7 brain networks? Does the visual system include some of dorsal and/or ventral attention areas? Is there an overlap in the brain regions?
3. Furthermore, it isn't clear if this analysis is across subjects or sessions, are the error bars std error across participants or sessions? Also is there are references for any of these fractions across brain areas or age? It would be useful to see actual values of these fractions across the adults and brain regions to get a sense of what these numbers are in addition to a repeatability measure. Do the numbers obtained in this study match a prior study? I think this would be a useful measure to test repeatability.

4. What does it mean the soma radius shows low repeatability compared to other measures? Is this a less reliable measure in that case? How do the numbers obtained in adults relate to numbers in children? Is there good repeatability of these metrics in during childhood as well?

5. It is unclear why the authors also measure cortical thickness and cortical volume. Do the authors have a specific hypothesis related to these anatomical measures and the microstructural measurements? If so, is there a relationship between thickness and each of the microstructural measures?

6. For analysis in Fig. 3 why did the authors choose the random forest model. Can the authors provide a sentence describing this analysis? Is it related to a correlation or regression model? Can you also add a table reporting the correlation R, p values per metric and brain areas? Does the relationship remain significant if you test for multiple comparisons? Can the author clarify if the reported R² is variance explained per region or mean across all regions, if across regions please provide std error.

7. It will be useful to conduct an ANOVA per metric with age, regions, and hemisphere as factors to see how significant these developments are across age, regions, or hemis?

8. I really like that the authors conducted a gene analysis to corroborate some of the outcomes of the MR analysis. However, I am not sure why the authors chose the DLPFC specifically when the prior analysis focused on the visual system and the one above that focused on larger brain networks. It will be beneficial to focus on the areas of the visual system if these samples are available in the postmortem dataset since the authors are validating the MR findings with histology (see Natu et al., 2021, Nature Communications Biology) else it is a challenge to expand results in the dlpc to a brain different region.

9. The authors also note that among the genes expressed in excitatory neuronal populations and oligodendrocytes, mean expression levels increased with age. Does this analysis age include infants? The analysis would be tighter if it would be limited to the age range within the 8-19 ages as in the MR data. Myelin development during infancy specifically is incredibly rapid and extensive in both gray and white matter and adding infants to your analysis can be misleading. I would then repeat the cell type analysis.

10. Can the authors provide a table listing the genes that are expressed in this analysis, prior to conducting the cell type analysis? What are the topmost differentially expressed genes during adolescence and do the authors find myelin-specific genes in their analysis? I guess I am wondering this because there are oligodendrocytes and OPCs in the cortex during adolescence and young adulthood, but they may or may not be actively myelinating at this time and I think this is important for the conclusion of the findings of this paper.

11. Minor comment: Can you please add units to Fig. 2d on the y-axis in addition to the one in the figure legend?

(Remarks on code availability)

Reviewer #2

(Remarks to the Author)

This study set out to examine the cortical microstructure during late childhood and adolescence (8-19 years). Specifically, the study utilized a unique MRI system with strong/fast gradients to acquire multi-shell diffusion MRI data for *in vivo* characterization paired with a gene expression analysis to evaluate the properties underlying cortical microstructural development. Biophysical modeling with the SANDI model was used to characterize cortical microstructure, while NODDI and DTI models were additionally fit as a comparison to SANDI and existing literature. Imaging analyses examined age-related associations of cortical microstructure from a sample of 88 participants, finding neurite signal fraction, intracellular volume fraction, and orientation dispersion index increased with age across examined cortical networks. The apparent soma radius and soma neurite signal decreased with age. Sex differences were also examined, with a limited number of networks displaying sex differences. Using a random forest model, the authors also evaluate the ability to predict age from cortical microstructural measures, with the apparent soma radius observed to provide the most accurate prediction. Using tissue samples available via two repositories (BrainCloud and PsychENCODE), the authors also modeled changes in normalized expression to identify genes differentially expressed over the studied age range. Results highlight that mean expression increased with age for genes expressed in excitatory neuronal populations and oligodendrocytes. Spatiotemporal patterns of gene expression additionally agree with patterns and timing of early myelination. Moreover, the spatial distribution of oligodendrocyte cell-type expression agreed with regional differences in the peak growth of the neurite fraction, suggesting that the neurite signal fraction and changes may reflect patterns of cortical myelination. Overall, I found the paper well written. The methods and results were well explained and the conclusions appear supported via the data. Combining the *in vivo* diffusion MRI data with gene expression data to better understand the underlying mechanisms is an exciting and novel approach and the results of these comparisons are interesting. However, while I found the paper of interest, I did have several comments:

1) To what extent is the scanner hardware (i.e. the ultra-strong gradients) enabling this research or improving the findings? Would this protocol be feasible on a clinical 3T system, and if so, what would be limitations of using these systems compared to the scanner used here?

2) The sex differences and age prediction analyses seemed to be a more supplemental analysis compared to the age

associations and comparison with gene expression age patterns. These also seem to be more disconnected with the rest of the paper and unclear why these were additionally included. The authors should try to provide more rationale for these more perhaps move to supplemental information?

3) Regarding sex differences, were such sex differences examined in the gene expression data?

4) The manuscript describes measuring DTI parameters from the $b=1000$ shell, however, in the description of the diffusion MRI protocol, there was no $b=1000$ shell. Was a separate or additional $b=1000$ shell acquired? For this DTI acquisition, were the same timing parameters as the multi-shell acquisition used? How many directions were acquired? If a separate acquisition was acquired with $b=1000$, presumably this would be fewer number of overall directions to the multi-shell protocol. How would this difference influence the comparisons with DTI?

5) How long was the overall diffusion MRI acquisition?

6) The methods describe corrections for susceptibility induced distortions. How were these performed? Were images with reverse phase encoding acquired?

7) For the age-related analyses, results from SANDI and NODDI models are reported, however, the manuscript also describes using DTI. It would be informative to include whether DTI parameters were also observed to be associated with age that was consistent with existing literature.

8) Were corrections for multiple statistical comparisons used?

(Remarks on code availability)

Code used in the manuscript seems to be available through other sources online, however, the authors do not specifically make their MRI processing code available. This would be a benefit to include.

Reviewer #3

(Remarks to the Author)

In this manuscript, Genc et al. examined the cross-sectional age related differences of the dMRI metrics among 88 individuals. Using their SANDI model, they estimated the cell fractions across brain regions and then perform associations with age. Simultaneously, they analyzed three different gene expression datasets (PsychENCODE, BrainCloud, Cell-type specific gene expressions), to infer what is the cell-type specific age-dependent curve of the gene expressions. The authors then compare those age-dependent curves obtained from dMRI and gene expressions, arguing that the best explanation for the change they observed is driven by the cortical myelinations.

In general, I found this paper intriguing and aligning with current trend on grounding the MRI metrics to the cell compositions. The authors put a tremendous effort on finding the common threads across datasets in completely different measurement modalities and scales, partly reflecting the limitations of current field, i.e. there is no common dataset that have all the variables/metrics necessary to answer the questions they have in mind. However, exactly because of this, the strength of this paper is also the weakness of the inference. If all the analyses dependent on one single variable, i.e. age, isn't that the estimated age curves would for sure to be similar? In a limited timeframe and resolution, the linear trend of the age can only go up, down, or stasis. How dMRI metrics can benefit the understanding while cortical thickness analyses already shown similar age dependent trends? What additional insight the SANDI based metrics bring while there were multiple reports with other type of dMRI metrics shown the age trajectories?

Besides the aforementioned main comment I have, here are some suggestions which might improve the accessibility of this paper to general readership of Nature Communications:

1. The initial analyses of the dMRI and the later gene expression analyses in the result section are not in the same ROIs. This create a cognitive dissonance about what exactly regions they are referring to.
2. The cell type specific trajectories are interesting but it puts a strong assumption on how gene expression changes over time in the bulk level. Better elaborate why this is a reasonable approach and empirically showcase the validity of such modeling.
3. The simulations in the final section of the result is very interesting. But it is unclear how it related to the gene expression curves they aim to compare with.
4. The authors have added some clinical implications in the discuss section, but those are speculative and have no direct evidence coming from their analyses.

(Remarks on code availability)

The URL contains multiple code repositories while the manuscript does not specify which one has been used. Therefore, it is impossible to review it.

Version 1:

Reviewer comments:

Reviewer #1

(Remarks to the Author)

The authors have addressed all my concerns and comments. The new figures and analyses strengthen the manuscript and its readability significantly. I believe this is impactful work and will make incremental progress in our field.

Reviewer #2

(Remarks to the Author)

The authors have done a commendable job of addressing the questions and concerns previously raised. I have no further questions and believe the manuscript is well written and findings provide new and noteworthy results.

Reviewer #3

(Remarks to the Author)

The authors have addressed all my concerns.

REVIEWER COMMENTS

We would like to thank the reviewers for their time in reviewing the current manuscript and helpful suggestions on improving the quality of the manuscript and interpretation of results. We also appreciate the positive comments regarding the strengths and novelty of the study.

We have revised the manuscript in accordance with the reviewer's recommendations, and believe the revised manuscript is considerably stronger with the proposed changes. We provide a point-by-point response to each of the reviewers' comments below. Where a change to the manuscript text was made, this specific text is represented in **red text**.

We welcome any additional feedback from the reviewers and thank them again for their time.

Reviewer #1 (Remarks to the Author):

Using ultra-strong gradient MRI to obtain high-resolution, in vivo estimates of cortical neurite and soma microstructure in typically developing children and adolescents, the authors examined the development of cortical microstructure and its underlying cellular mechanisms. The authors report that the neurite signal fraction, attributed to neuronal and glial processes increased with age whereas the soma radius decreased with age across brain networks. To complement the findings, the authors also examined developmental patterns of cortical gene expression in two independent post-mortem databases. Results showed increased expression of genes expressed in oligodendrocytes, and excitatory neurons, and a decrease in expression of genes expressed in astrocyte, microglia and endothelial cell-types in adolescence and young adulthood. The authors report that the developmental patterns in cortical neurite and soma highlight the contribution of active myelination processes in cortical maturation during adolescence.

The authors are attempting to answer a very important question in neuroscientific literature regarding the maturation of cortical microstructure during a critical period of human development. Further, they are investigating the development using a multiple model approach, combining ultra-strong gradient imaging and as well as transcriptomics, which is very exciting, as combining different methods can allow us to dig deeper into the underlying cellular and molecular mechanisms making impactful discoveries in the field of developmental neuroscience.

That said, I have a few comments, queries, and clarification on the methods and analysis that the authors used in the current study. I specifically found the analysis lacking a logical flow and confusing. I think the authors are analyzing several morphological and microstructural metrics which are important, however, there is a lack of clear reasoning of why the authors selected these metrics, or brain areas, and opens several concerns which I list below:

Methodological concerns.

1. Concerns with Figure 1: I really like the schematic in Fig. 1, however, it needs color bars telling the reader what the brain maps indicate, what do the warmer colors indicate in each map? What is the range of numbers per metric and what do they indicate exactly? The authors are using novel methods and metrics to examine cortical development, and it would

be beneficial for the reader to know the range of fractions for $f_{neurite}$, f_{soma} , and $f_{extracellular}$ and radius range for R_{soma} within a voxel. Additionally, a paragraph on how these fractions /measures are obtained will be helpful and what information each measure provides will be also helpful. I would also show a comparison here between the 8-year-old sample child and an adult from one of 6 adults to help the reader understand where and what the development looks like on each of these maps.

Thank you for your feedback in improving the interpretability of our methods and results. To aid in interpreting the colour bars for two participants, we have generated a new figure that repeats the four maps in Figure 1 (for a representative 8-year-old) and additionally shows the same measures for a representative 30-year-old participant, using the same colour scale. We have created a new figure to illustrate this point, as to avoid overcrowding the workflow schematic in Figure 1.

Figure S11: Representative maps of the four SANDI measures from an 8-year-old and a 30-year-old.

We agree that some further information on the interpretation of SANDI measures will help translate these findings to a wider audience. To this point we have added a paragraph to the introduction to clarify the interpretations of these measures in the cortical grey matter (line 73).

"SANDI is a biophysical tissue model that estimates the diffusion-weighted signal contribution from three distinct compartments: intra-neurite, intra-soma, and extracellular space. For each imaging voxel, a signal fraction will be estimated for each of the three compartments, such that they sum to 1. In the cortical grey matter, there is a higher proportion of soma (neuronal and glial cell bodies) to neurites, leading to a higher soma signal fraction. These signal fractions vary around tissue boundaries, with higher extracellular signal fraction around the cortical surface due to partial voluming with CSF. Overall, these compartment-specific signal fractions are relative, and comparing

these trends over age are potentially meaningful to deduce the compartments that are contributing most to age-related changes in cortical development."

2. In the first analysis, the authors examined the repeatability of cortical microstructural estimates from the SANDI model in a sample of 6 healthy adults scanned over 5 sessions. This is a useful analysis, but it is unclear why the authors choose to report an intra-class coefficient (ICCs) per neurite signal fraction (fneurite), soma signal fraction (fsoma) and extracellular signal fraction (fextracellular) metric and how this is calculated and why the authors chose the 7 brain networks? Does the visual system include some of dorsal and/or ventral attention areas? Is there an overlap in the brain regions?

Thank you for your question. As SANDI measures have not been widely investigated since its recent inception, and to the best of our knowledge no study has reported the test-re-test reliability of such measures using a moderate (lower b-value) acquisition regime, we thought it was a useful analysis to speak to the reliability and stability of each of the individual measures. In the absence of paediatric repeatability data, we looked at a group of adults who were scanned on the same MRI system as the developmental cohort, with the same acquisition protocol. By reporting the ICC for each SANDI measure, we can gain more confidence in our ability to detect meaningful developmental patterns. Our results show that SANDI measures are highly stable across different brain regions, reflected by the high ICC values (mean ICC=0.95) across all regions and measures studied. We provide more detail on how the ICC is calculated in the response to Q3 below.

We specifically chose a functional atlas for this study as there are limited structural atlases that provide detailed parcellation of the brain beyond the four main lobes (frontal, parietal, temporal, occipital). These anatomically defined lobes may not capture the fine-grained variations in microstructure across functionally distinct areas. By using a functional atlas we may be better positioned to make conclusions around developmental patterns of neural activity and connectivity (Fotiadis et al., 2024).

Line 456: *"We chose a functional atlas due to the limitations of structural atlases in capturing fine-grained microstructural variations, enabling better insights into developmental patterns of neural activity and connectivity (Fotiadis et al., 2024)."*

The derivation of the functional networks in subject space was performed as previously described in Genc et al. (2024). Briefly, we co-registered the Yeo functional atlas (Yeo et al., 2011) in MNI space to each individual subject's space using a non-linear transformation to obtain seven functionally relevant cortical canonical networks (visual, somatomotor, dorsal attention, ventral attention, limbic, frontoparietal, default mode network). The majority of statistical analyses were performed using voxel-averaged values within each volumetric mask of the seven networks.

For the age-prediction analysis in Figure S10, we wanted to identify specific regions commonly belonging to the visual network. Following the method described in Genc et al. (2024), we grouped HCP-MMP1 parcels overlapping in at least 80% of the enrolled subjects, discarding any ROIs that did not meet these criteria. This resulted in regions that were uniquely represented in the visual network. To generate the prefrontal cortex overlap as described in Fig 4, we used the regions defined in Glasser et al. 2016 ("Supplementary Neuroanatomical Results").

To clarify these points further in the current manuscript, we have provided more details in the methods (section 4.1.2; line 453), as follows:

"The network-level atlas derivation is detailed in Genc et al. (2024). Briefly, we co-registered the Yeo functional atlas in MNI space to each individual subject's space, to obtain seven functionally relevant cortical canonical networks (visual, somatomotor, dorsal attention, ventral attention, limbic, frontoparietal, default mode network)."

3. Furthermore, it isn't clear if this analysis is across subjects or sessions, are the error bars std error across participants or sessions? Also is there are references for any of these fractions across brain areas or age? It would be useful to see actual values of these fractions across the adults and brain regions to get a sense of what these numbers are in addition to a repeatability measure. Do the numbers obtained in this study match a prior study? I think this would be a useful measure to test repeatability.

Apologies for the confusion, we computed the intra-class correlation coefficient using two-way random effects, absolute agreement, which summarises the agreement across sessions within individual subjects. We used the 'ICC' function from the 'psych' package in R. We input the SANDI measures averaged across entire networks and treat both subjects and sessions as random effects. The error bars represent the bounds of the 95% confidence interval (CI) for each ICC estimate. We have added more detail to the methods (section 4.3.1, line 491) to describe the repeatability analysis, as follows:

"For the repeatability analysis, the intra-class correlation coefficient (ICC; two-way random effects, absolute agreement) was computed for assessment of test-re-test repeatability for SANDI and DTI metrics using the 'psych' package in R. Lower and upper estimates of each ICC represent the bounds of the 95% confidence interval (CI)."

We have plotted the adult data alongside the developmental cohort to highlight the magnitude of values and the trend of imaging measures with age (Figure R1, below). We also append a table below (see response to next reviewer question; Table R1) with mean and median absolute deviation of imaging measures.

Figure R1: Full span of DTI and SANDI data from both cohorts to capture developmental (N=88) and adult data from one session (N=6) plotted on the same curve.

4. What does it mean the soma radius shows low repeatability compared to other measures? Is this a less reliable measure in that case? How do the numbers obtained in adults relate to numbers in children? Is there good repeatability of these metrics in during childhood as well?

The ICC of Rsoma was very high (ICC > 0.93; range = 0.953 – 0.985) for 6/7 networks (see table below). This speaks to the high intra-subject repeatability of the Rsoma measure derived from the SANDI model. In the limbic network, the ICC of Rsoma was moderate (ICC=0.66), which resulted in our reported “average ICC” to be lower than for the other SANDI metrics (mean ICC=0.92). Therefore, we would stipulate that estimates of Rsoma would be less reliable in the limbic network, likely due to susceptibility artefacts proximal to fronto-temporal regions.

Table R1: Mean, median absolute deviation (MAD) and intra-class correlation coefficients (ICC) of SANDI measures in the adult repeatability dataset.

	f _{extracellular}				f _{neurite}			
	Mean	MAD	ICC	p val	Mean	MAD	ICC	p val
Visual	.41	.014	.984	< .001	.14	.008	.950	< .001

Somatomotor	.45	.031	.995	< .001	.12	.007	.954	< .001
Dorsal attention	.44	.029	.997	< .001	.11	.008	.962	< .001
Ventral attention	.43	.023	.987	< .001	.11	.010	.965	< .001
Limbic	.41	.015	.935	< .001	.18	.012	.962	< .001
Fronto-parietal	.45	.033	.995	< .001	.11	.004	.934	< .001
Default	.44	.023	.992	< .001	.11	.005	.965	< .001

	f_{soma}				$R_{soma} (\mu m)$			
	Mean	MAD	ICC	pval	Mean	MAD	ICC	pval
Visual	.46	.017	.985	< .001	8.81	.110	.934	< .001
Somatomotor	.43	.033	.997	< .001	8.75	.130	.983	< .001
Dorsal attention	.45	.033	.996	< .001	8.81	.130	.985	< .001
Ventral attention	.46	.018	.991	< .001	8.90	.130	.939	< .001
Limbic	.42	.015	.891	< .001	8.68	.040	.656	.04
Fronto-parietal	.45	.024	.996	< .001	8.86	.130	.974	< .001
Default	.45	.020	.996	< .001	8.84	.130	.962	< .001

Abbreviations: ICC = Intra-class correlation; MAD = median absolute deviation

To aid in interpreting the absolute values derived from our dataset, we now compare our data to a recently published paper that studied these SANDI measures in a sample of adults aged 19-85 years of age (Lee et al., 2024).

Our reported soma signal fraction is 0.45 vs 0.43 (Lee et al., 2024), the extracellular signal fraction is 0.43 vs 0.40, the neurite signal fraction is 0.13 vs 0.16 and the soma radius is 8.8 vs 9.3 μm . We also note that the values from Lee et al. are averaged across the entire cerebral cortex, whereas our data is averaged across the seven networks, which may result in some regions being overrepresented and with a larger volume, potentially influencing mean estimates. Overall, our findings from our repeatability component of our study are comparable with this recently published study using SANDI.

From Lee et al. (2024), (Supplementary table 1):

	All Participants N = 72	Young (under 34) N = 33	Middle (35 to 54) N = 21	Older (over 55) N = 18
f_{is}	0.42 \pm 0.02	0.43 \pm 0.01	0.41 \pm 0.02	0.40 \pm 0.01
f_{in}	0.17 \pm 0.01	0.16 \pm 0.01	0.17 \pm 0.01	0.17 \pm 0.01
f_{ec}	0.41 \pm 0.02	0.40 \pm 0.02	0.41 \pm 0.02	0.43 \pm 0.02
$r_s (m)$	9.26 \pm 0.14	9.30 \pm 0.16	9.28 \pm 0.08	9.17 \pm 0.11

No study, to the best of our knowledge, has employed these comparisons in a developmental cohort. As further described in the response to Q2, we did not acquire repeatability data in children, and we are not aware of any repeatability datasets acquired in children with an acquisition scheme suitable for SANDI modelling. It is challenging to acquire such data in paediatric populations, firstly

due to the frequency of scans needed within a short window, and secondly due to the rapid nature of brain development which may pose difficulty in making conclusions around test-re-test reliability of measures that can change within the timescale of repeated scans (weeks or months). However, we would expect similar patterns of high repeatability across all networks due to the consistently high ICC values observed in our sample of 6 adults aged 24-30 years of age.

We have described this further in the strengths and limitations section of the manuscript (Line 368):

"Our repeatability results show that SANDI-derived biophysical signal fractions are highly stable (mean ICC=.97) in a young adult population, and these values are highly concordant with recently reported cortex-wide measurements in a subset of younger adults (Lee et al., 2024)."

5. It is unclear why the authors also measure cortical thickness and cortical volume. Do the authors have a specific hypothesis related to these anatomical measures and the microstructural measurements? if so, is there a relationship between thickness and each of the microstructural measures?

Thank you for this important question. Indeed, many developmental studies consistently report reductions in cortical thickness and cortical volume across childhood and adolescence (Mills 2016; Tamnes 2017; Giedd 1999). The typical interpretation around these observations has been synaptic pruning – however there is no way to directly measure such a fine-scale parameter using macrostructural T1-weighted MRI. dMRI can get us closer to estimating the signal fraction arising from different tissue compartments, based on the principles of water diffusion. As opposed to macrostructural measures such as thickness and volume, dMRI is uniquely positioned to reveal the microstructural underpinning of changes in brain development, or even detect microstructural changes in the absence of net macrostructural change.

It is an interesting experiment to relate measures of cortical morphology with individual diffusion measures to understand the substrate of reported differences in morphology. We briefly report some partial correlations for cortical thickness (CTh), grey matter volume (GMvol) and surface area (SA) below:

```
Model 1: CTh_[network1:7] <- lm(CTh ~ age + fe + fneurite + fsoma + rsoma+ sex  
, data=data_yeo7networks_[network1:7])
```

```
Model 2: GMvol_[network1:7] <- lm(GMvol ~ age + fe + fneurite + fsoma + rsoma+ sex  
, data=data_yeo7networks_[network1:7])
```

```
Model 3: SA_[network1:7] <- lm(SA ~ age + fe + fneurite + fsoma + rsoma+ sex ,  
data=data_yeo7networks_[network1:7])
```

Figure R2: Results from the partial correlations for cortical thickness, grey matter volume and surface area. Results show the percentage variance of each of the model terms, which included all four SANDI measures as well as age and sex.

The high residuals in each of the statistical tests (Fig R2) suggests that a large proportion of variance in cortical morphology (thickness, volume, surface area) is not explained by the microstructural measures. This highlights the potentially unique information that microstructural measures are providing around developmental differences, above and beyond trends in cortical morphology. It is also possible that the remaining variance is due to the myelin content itself, however dMRI is relatively insensitive to the myelin signal, as the signal from myelin has decayed away beyond detection in typical dMRI sequences. As such we cannot make direct conclusions around the contributions of myelin content to these observed patterns.

Few measures showed higher contributions of microstructural measures to morphology estimates. For example, Rsoma contributed to 8.5% of variance in CTh in the ventral attention network, and 7.8% of variance in surface area in the default mode network. Figure S7 also shows the peak maturation of microstructural measures and cortical thickness for comparison. Here, we observe that peak maturation of CTh occurred earlier in the dorsal attention, visual and default mode networks. These spatiotemporal patterns of peak maturation most closely resembled those observed in the extracellular signal fraction.

Overall, our results show that cortical microstructural measures provide unique information around cortical development, unique to that estimated by measures of cortical morphology.

6. For analysis in Fig. 3 why did the authors choose the random forest model. Can the authors provide a sentence describing this analysis? Is it related to a correlation or regression model? Can you also add a table reporting the correlation R, p values per metric and brain areas? Does the relationship remain significant if you test for multiple comparisons? Can the author clarify if the reported R² is variance explained per region or mean across all regions, if across regions please provide std error.

We used a random forest (RF) model instead of standard regression due to its ability to model nonlinear relationships, shorter training times, reduced risk of overfitting, and interpretability of feature importance. Specifically, the depth at which a feature appears as a decision node in a tree provides insight into its relative significance for predicting the target variable (in this case, age). Features located near the top of the tree influence the prediction outcome for a larger portion of the input samples. This allows us to assess the relative contribution of each brain region collectively rather than conducting separate analyses per region, providing insights into feature importance without the need for individual statistical tests.

In response to the reviewer’s request, we examined the statistical significance of the age prediction model (correlation of predicted vs. actual age) for each of the five measures presented in Fig. 3 (now Fig. S10) on the holdout set (N=18). All measures remain significant after multiple comparisons correction ($p < .005$) and pass Bonferroni correction ($0.05/5=0.01$).

Table R2: Statistical testing of the age prediction models.

Measure	R ²	p-value
Rsoma	.58	3.38e-05
fsoma	.28	0.002
fneurite	.56	3.93e-05
odi	.46	3.6e-04
vic	.36	0.003

We have clarified this point further in the methods (Line 507):

“We chose a RF model due their ability to model nonlinear relationships, reduced risk of overfitting, and interpretability of feature importance. Specifically, the depth at which a feature appears as a decision node in a tree provides insight into its relative significance for predicting the target variable (i.e., age). This allows us to assess the relative contribution of features (i.e., average signal fraction in each HCPMMP1 parcel) to age prediction.”

7. It will be useful to conduct an ANOVA per metric with age, regions, and hemisphere as factors to see how significant these developments are across age, regions, or hemis?

To answer these questions, we have now run the following models:

Model 1: `summary(aov(fneurite ~ age + network + Error(ID/network), data=data_yeo7networks))`

Model 2: `summary(aov(fsoma ~ age + network + Error(ID/network), data=data_yeo7networks))`

Model 3: `summary(aov(fe ~ age + network + Error(ID/network), data=data_yeo7networks))`

Model 4: `summary(aov(rsoma ~ age + network + Error(ID/network), data=data_yeo7networks))`

Below are the results for each model:

	F(age)	p(age)	F(network)	p(network)
fneurite	156.3	<2e-16	585.9	<2e-16
fsoma	6.63	0.0117	485.2	<2e-16
fextracellular	12.74	0.000588	196.3	<2e-16

Rsoma	143.1	<2e-16	349.7	<2e-16
-------	-------	--------	-------	--------

As these measures were estimated at the whole network level, to investigate the effect of hemisphere, we performed statistical testing of SANDI measures in HCPMMP1 parcels derived for the visual network (for the age prediction analysis presented).

```
summary(aov(value ~ age + hemisphere + variable + Error(1+ID), data=hcp_fneurite_long))
summary(aov(value ~ age + hemisphere + variable + Error(1+ID), data=hcp_fsoma_long))
summary(aov(value ~ age + hemisphere + variable + Error(1+ID), data=hcp_fe_long))
summary(aov(value ~ age + hemisphere + variable + Error(1+ID), data=hcp_rsoma_long))
```

Visual network						
	F(age)	p(age)	F(region)	p(region)	F(hemisphere)	p(hemisphere)
fneurite	118.9	<2e-16	128.72	<2e-16	59.38	1.47E-14
fsoma	2.342	0.13	86.67	<2e-16	48.87	2.98E-12
fextracellular	19.57	2.83E-05	44.43	<2e-16	188.63	<2e-16
Rsoma	86.67	1.15E-14	81.78	<2e-16	36.65	1.48E-09

These results in the visual network show that each SANDI measure significantly varies between regions that make up the network ($p < 2e-16$) and between hemispheres ($p < 1.5e-9$).

To visualise these differences, we have plotted the relationship between age and SANDI metrics and grouped the data by hemisphere. We observe that fneurite and fsoma are higher in the right hemisphere throughout development, and Rsoma and fextracellular are higher in the left hemisphere throughout development.

Figure R3: Relationship between age and diffusion metrics between the left and right hemispheres.

8. I really like that the authors conducted a gene analysis to corroborate some of the outcomes of the MR analysis. However, I am not sure why the authors chose the DLPFC specifically when the prior analysis focused on the visual system and the one above that focused on larger brain networks. It will be beneficial to focus on the areas of the visual system if these samples are available in the postmortem dataset since the authors are validating the MR findings with histology (see Natu et al., 2021, Nature Communications Biology) else it is a challenge to expand results in the dlpc to a brain different region.

Thank you for allowing us to clarify this analysis further. We fit the model using data for all regions, but we display gene expression data from the DLPFC to allow for clearer comparisons of

developmental trajectories between BrainCloud and PsychENCODE cohorts (Figure 3a,b). Whilst the BrainCloud sample is a rich dataset with 214 post-mortem brains, it only includes samples from the DLPFC, so our decision to restrict our visualisation to the DLPFC in Figure 3a,b was purely for direct comparison with PsychENCODE data. The developmental trajectories for the remaining brain regions in PsychENCODE are shown in Figure 3g. It can be appreciated in this figure that the maturation of V1 is much earlier than other association and frontal regions, which we elaborate on in the Discussion in terms of earlier timing of cortical myelination in primary visual and motor cortices.

We have clarified this further in line 176:

“Developmental profiles from the DLPFC were visualised, to allow for clearer comparisons of cell-type specific trends over age between cohorts.”

9. The authors also note that among the genes expressed in excitatory neuronal populations and oligodendrocytes, mean expression levels increased with age. Does this analysis age include infants? The analysis would be tighter if it would be limited to the age range within the 8-19 ages as in the MR data. Myelin development during infancy specifically is incredibly rapid and extensive in both gray and white matter and adding infants to your analysis can be misleading. I would then repeat the cell type analysis.

Thank you for raising an important point regarding expected myelination patterns throughout the lifespan. We chose to include the full age span from 6 months of age until 40 years of life, as we expected to see mean expression levels increasing up to mid-life. We wanted to maximise our age range to maximise the amount of variance of gene expression over age to better capture non-linear trajectories. In a smaller age window, we expect mean expression levels to either go up, go down or stay the same. By including a larger age range, we were able to identify any genes that change over time non-linearly.

In Figure 4 we see that the expression of oligodendrocyte cell-types are increasing throughout the full window included. If these genes were only involved in early myelination – we would expect to see a flattening of this expression curve following infancy. Similarly, if there was a “lull” in oligodendrocyte cell-type gene expression between early childhood and adolescence, we would expect to see a flattening of the curve in this developmental period.

However, taking into account the reviewer’s concerns, we agree with the point that we may be more sensitive to genes specifically related to cortical myelination if we studied a tighter age range precluding the early postnatal and infancy period, a time where white matter myelination is known to occur rapidly. Upon reanalysis of the gene expression data excluding samples less than 8 years of age, we observe N=155 genes that significantly increase with development across the two datasets (PsychENCODE and BrainCloud). All of these genes were included in the original geneset (33% of N=467 genes that significantly increase over age 0.5-30 years). The genes in this reduced set of N=155 still include genes involved in oligodendrocyte expression identified in the original analysis (e.g., RCAN2, PRUNE2).

To strengthen our conclusions even further around expression patterns across child and adolescent development, we took advantage of recently published single-cell RNA data (Velmeshev et al., 2023) and performed an additional validation - please see the response to Reviewer 3, Q2. Overall, we are

confident that the increases we see are not dominated by early postnatal period of myelination and that increases in oligodendrocyte cell-type expression persists throughout childhood and adolescence.

10. Can the authors provide a table listing the genes that are expressed in this analysis, prior to conducting the cell type analysis? What are the topmost differentially expressed genes during adolescence and do the authors find myelin-specific genes in their analysis? I guess I am wondering this because there are oligodendrocytes and OPCs in the cortex during adolescence and young adulthood, but they may or may not be actively myelinating at this time and I think this is important for the conclusion of the findings of this paper.

We report the genes that were included in the analysis in the Supplementary information (section 8.3). Out of the 467 genes that showed significant age-related development, 406 of these genes could be mapped to specific cell types (Figure S5) and 446 were mapped to specific developmental windows (Figure S6). The genes in supplementary section 8.3 are genes with significant age effects, prior to conducting the cell type analysis.

As per the reviewer's request, we list the genes that were significantly enriched in adolescent and young adulthood in the CSEA, as follows:

Cortex.Adolescence: RCAN2, FAM153B, EGR2, EGR1, ARHGAP25, PI4KA, GLS, NECAB1, KIAA0513, PIK3R1, NAPEPLD, PLEKHA1, SCN1B, KIF5A, ABLIM2, VSNL1, SCN1A, SPARCL1, SNAP25, RASGRF2, KCNAB2, SORL1, RIMS3, TNNT2, MCF2L2, TPPP, HSD11B1, CAMK2G, STAMBPL1, BHLHE40, GFOD1, IQSEC1, CORO6, ETS2, SYT12, OXR1, EPHB6, ENPP5, CABP1, DNM1, CAMKK1, CX3CL1

Cortex.Young.Adulthood: CAMK1D, RCAN2, FAM153B, TSPYL2, MPP1, VWA5B2, PTPRT, FAM81A, PI4KA, GLS, ABCC8, ARHGAP26, NECAB1, GABRG2, PLEKHA6, COL24A1, KIAA0513, PIK3R1, NAP1L2, GLDN, ITPR1, RAPGEF2, NAPEPLD, PTPRK, AATK, SCN1B, PLEKHA1, HHATL, KIF5A, ABLIM2, VSNL1, SCN1A, KCNC1, SPARCL1, SORT1, CPLX1, SNAP25, RASGRF2, KCNAB2, SORL1, RIMS3, DNM1, TNNT2, SYNJ2, MCF2L2, MFSD6, TPPP, TMEM130, HSD11B1, CAMK2G, STAMBPL1, TTBK2, BHLHE40, GFOD1, IQSEC1, CORO6, ETS2, SYT12, OXR1, EPHB6, PRUNE2, DOCK5, CABP1, GRIA3, CAMKK1, SIPA1L1

The nature of the data we have used means we cannot identify cell-type specific indicators of gene expression, however, from our gene list we identified several genes involved in myelination such as RCAN2 and GRIA3, as well as genes involved in the differentiation of OPCs and oligodendrocytes, PLEHA1/TAPP1 and AATK/AATYK. Genes such as RCAN2 and PRUNE2 still showed significant age-related increases in expression even when mapping trajectories from age 8 years onwards (see also response to Reviewer 3, Q2). We have discussed this further in the scRNA validation and made modifications to the text accordingly.

11. Minor comment: Can you please add units to Fig. 2d on the y-axis in addition to the one in the figure legend?

We have now added units to Figure 2d in the y-axis.

Reviewer #2 (Remarks to the Author):

This study set out to examine the cortical microstructure during late childhood and adolescence (8-19 years). Specifically, the study utilized a unique MRI system with strong/fast gradients to acquire multi-shell diffusion MRI data for in vivo characterization paired with a gene expression analysis to evaluate the properties underlying cortical microstructural development. Biophysical modeling with the SANDI model was used to characterize cortical microstructure, while NODDI and DTI models were additionally fit as a comparison to SANDI and existing literature. Imaging analyses examined age-related associations of cortical microstructure from a sample of 88 participants, finding that neurite signal fraction, intracellular volume fraction, and orientation dispersion index increased with age across examined cortical networks. The apparent soma radius and soma neurite signal decreased with age. Sex differences were also examined, with a limited number of networks displaying sex differences. Using a random forest model, the authors also evaluate the ability to predict age from cortical microstructural measures, with the apparent soma radius observed to provide the most accurate prediction. Using tissue samples available via two repositories (BrainCloud and PsychENCODE), the authors also modeled changes in normalized expression to identify genes differentially expressed over the studied age range. Results highlight that mean expression increased with age for genes expressed in excitatory neuronal populations and oligodendrocytes. Spatiotemporal patterns of gene expression additionally agree with patterns and timing of early myelination. Moreover, the spatial distribution of oligodendrocyte cell-type expression agreed with regional differences in the peak growth of the neurite fraction, suggesting that the neurite signal fraction and changes may reflect patterns of cortical myelination. Overall, I found the paper well written. The methods and results were well explained and the conclusions appear supported via the data. Combining the in-vivo diffusion MRI data with gene expression data to better understand the underlying mechanisms is an exciting and novel approach and the results of these comparisons are interesting. However, while I found the paper of interest, I did have several comments:

1) To what extent is the scanner hardware (i.e. the ultra-strong gradients) enabling this research or improving the findings? Would this protocol be feasible on a clinical 3T system, and if so, what would be the limitations of using these systems compared to the scanner used here?

We thank the reviewer for raising this important question. Below, we provide a twofold response, firstly to clarify the role of the Connectom scanner's ultra-strong gradients and secondly to discuss the feasibility of adapting the protocol for clinical 3T systems.

1. Positioning of Our Study

Our study was aimed to address a highly specific question in the neuroscientific literature regarding cortical microstructural maturation during a critical developmental period. This investigation leverages the unique capabilities of the Connectom scanner to combine ultra-strong gradient imaging with transcriptomics, enabling us to explore cortical microstructure with greater sensitivity and specificity than would be achievable with a standard clinical scanner. We emphasize that the Connectom scanner is an experimental platform designed for cutting-edge research and is not representative of hardware available in most neuroimaging facilities.

We recognize that translating such protocols for use on standard clinical 3T systems is crucial for facilitating similar analyses in other settings. Recent studies have implemented similar protocols on a commonly available clinical 3T scanner (Siemens Magnetom Prisma) with a gradient amplitude of 80 mT/m, which we had previously outlined in the “Clinical Implications” section (now renamed “Potential Applications”) of our manuscript (see Barakovic et al., 2024; Margoni et al., 2023; Schiavi et al., 2023). Broadly, several changes to the protocol used here are necessary to accommodate the hardware limitations.

2. Feasibility and Limitations on Clinical Systems

Key differences between the Connectom and Prisma implementations of the SANDI protocol are summarized below:

- *Gradient Pulse Duration (δ):* On the Connectom, the minimum gradient pulse duration that allows us to achieve high b-values (6000 s/mm^2) with short diffusion times ($\Delta = 24 \text{ ms}$) was 7 ms. In contrast, to achieve the same maximum b-value, the Prisma requires a longer gradient pulse duration due to its lower maximum gradient amplitude, which in turn increases the diffusion time. This has potential implications on the interpretability of findings in the cortical grey matter due to exchange.
- *Echo Time (TE):* As a result of the shorter gradient pulse duration, the Connectom allows for a shorter TE, minimizing T_2 -related signal loss and maximizing SNR per unit time. On the Prisma, TE is necessarily extended and potentially affects the sensitivity to microstructural features.

These trade-offs highlight the value of the Connectom scanner for addressing highly specific research questions, as in our study. Nonetheless, adapting protocols for more accessible scanners remains a priority. Recently, commercially available systems can overcome these limitations and they are expected to increasingly replace the current state-of-the-art clinical grade 3T MRI systems. Such examples of these platforms include the Siemens 3T Cima.X scanner, and the GE 3T MAGNUS scanner, both with a $G_{\max} > 200 \text{ mT/m}$.

We have clarified the benefits of the Connectom scanner as follows:

Line 355: *“One potential future application is to quantify cortical microstructure in such clinical cohorts, especially with adaptations towards clinically feasible acquisition protocols using current state-of-the-art clinical grade 3T systems (Barakovic et al., 2024; Margoni et al., 2023; Schiavi et al., 2023), and with the recent advent of commercial systems with ultra-strong gradients (e.g., Siemens 3T Cima.X; GE 3T MAGNUS).”*

Line 371: *“Using in vivo microstructural imaging with ultra-strong gradients ($G_{\max}=300 \text{ mT/m}$; Jones et al. (2018)), we achieved sensitivity to micrometer-level imaging contrast by maximising SNR and minimising the effect of water exchange (Raven et al., 2023).”*

2) The sex differences and age prediction analyses seemed to be a more supplemental analysis compared to the age associations and comparison with gene expression age patterns. These also seem to be more disconnected with the rest of the paper and unclear why these

were additionally included. The authors should try to provide more rationale for these more perhaps move to supplemental information?

We agree that this analysis would fit better in the supplementary section, as such we have moved the Figure 3 (now Figure S10) and the results into the supplementary section. This allows us to focus the main findings of the paper around the concordance of developmental patterns of microstructure and gene expression.

3) Regarding sex differences, were such sex differences examined in the gene expression data?

Given the conservative sample size (e.g., $n=10$ in PsychENCODE data), we did not investigate sex differences in the gene expression data. We agree this is important for future research with larger sample sizes to investigate.

4) The manuscript describes measuring DTI parameters from the $b=1000$ shell, however, in the description of the diffusion MRI protocol, there was no $b=1000$ shell. Was a separate or additional $b=1000$ shell acquired? For this DTI acquisition, were the same timing parameters as the multi-shell acquisition used? How many directions were acquired? If a separate acquisition was acquired with $b=1000$, presumably this would be fewer number of overall directions to the multi-shell protocol. How would this difference influence the comparisons with DTI?

Apologies, this was an error in the methods description as it should read that " $b=1200$ " data were used for the DTI fit. We have amended the methods (section 4.1.2, line 448) to reflect this, as follows:

"...(DTI) metrics were estimated using the $b=1200$ s/mm² shell (Fractional anisotropy (FA); mean diffusivity (MD, in s/mm²))."

And in the results (section 2.1, line 143):

"We observed sex differences in only two microstructural measures, R_{soma} and fractional anisotropy (FA; derived from the diffusion tensor at $b=1200$ s/mm²), in the visual network (Fig S2, S3)."

5) How long was the overall diffusion MRI acquisition?

The total acquisition time was 16 minutes and 14 seconds, acquired across four acquisition blocks to improve compliance in paediatric populations, as reported in Genc et al. (2020). We have amended the methods (section 4.1.2, line 430) to reflect this, as follows:

"The total acquisition time (across four acquisition blocks) was 16 min 14 s."

6) The methods describe corrections for susceptibility induced distortions. How were these performed? Were images with reverse phase encoding acquired?

Our apologies for omitting these details from the methods. We acquired one volume with no diffusion-weighting in the reverse phase encoding direction to perform susceptibility-induced

distortion correction interfacing tools in FSL and MRtrix3. We have added the following description to the methods (line 429):

"Data were acquired in an anterior–posterior (AP) phase-encoding direction, with one additional PA volume."

7) For the age-related analyses, results from SANDI and NODDI models are reported, however, the manuscript also describes using DTI. It would be informative to include whether DTI parameters were also observed to be associated with age that was consistent with existing literature.

Thank you for the suggestion. We have previously reported these age associations in Supplementary Figure S1 and Table S2, however we have now added these to the main text (line 125) for ease of interpretability and comparison with existing literature.

"DTI metrics revealed decreasing FA with age across all networks apart from the limbic network (mean $R_2 = .27$, $p < 2.1e-5$), decreasing MD in the limbic network, $\beta = -.42$, $[-.62, -.22]$, $p = 8.8e-5$, and increasing MD in the somatomotor network, $\beta = .34$, $[.13, .55]$, $p = .002$."

8) Were corrections for multiple statistical comparisons used?

We chose to set a strict p-value threshold of $p < .005$ (as per recommendations in Benjamin et al, Nature Human Behaviour, 2018), rather than multiple comparisons correction, to improve the reproducibility of our results.

Reviewer #2 (Remarks on code availability):

Code used in the manuscript seems to be available through other sources online, however, the authors do not specifically make their MRI processing code available. This would be a benefit to include.

Thank you for the suggestion, we have now included the MRI processing code for review.

Reviewer #3 (Remarks to the Author):

In this manuscript, Genc et al. examined the cross-sectional age related differences of the dMRI metrics among 88 individuals. Using their SANDI model, they estimated the cell fractions across brain regions and then perform associations with age. Simultaneously, they analyzed three different gene expression datasets (PsychENCODE, BrainCloud, Cell-type specific gene expressions), to infer what is the cell-type specific age-dependent curve of the gene expressions. The authors then compare those age-dependent curves obtained from dMRI and gene expressions, arguing that the best explanation for the change they observed is driven by the cortical myelinations.

In general, I found this paper intriguing and aligning with current trend on grounding the MRI metrics to the cell compositions. The authors put a tremendous effort on finding the common threads across datasets in completely different measurement modalities and scales,

partly reflecting the limitations of current field, i.e. there is no common dataset that have all the variables/metrics necessary to answer the questions they have in mind. However, exactly because of this, the strength of this paper is also the weakness of the inference. If all the analyses dependent on one single variable, i.e. age, isn't that the estimated age curves would for sure to be similar? In a limited timeframe and resolution, the linear trend of the age can only go up, down, or stasis. How dMRI metrics can benefit the understanding while cortical thickness analyses already shown similar age dependent trends? What additional insight the SANDI based metrics bring while there were multiple reports with other type of dMRI metrics shown the age trajectories?

We thank the reviewer for their carefully considered comments. There are several aspects of our study that help to improve the status quo of understanding the developing brain through childhood and adolescence. Firstly, and most importantly, this study is the first attempt at robust modelling that properly accounts for all cellular structures to study maturation of the cortex. Rather than applying approaches (such as DTI) that were developed for the white matter, we report for the first time in a developmental cohort, the application of a biophysical model specifically designed to characterise cortical tissue microstructure. By doing this, we can go above and beyond what is known about developmental patterns of cortical thickness (a coarse, macrostructural measure) and attempt to disentangle various compartmental contributions (i.e. separating axon from cell body) which can be conflated in measures of thickness.

Up until recently, most studies into microstructure of the cortex have used simple DTI measures such as MD. This can be very informative and sensitive to various pathological alterations to tissue microstructure, however, MD is simply an average of water diffusion in three directions ($\lambda_1, \lambda_2, \lambda_3$) of the diffusion tensor. Since DTI is not a model but a representation of the diffusion-weighted signal (Novikov DS, et. al., Magnetic Resonance in Medicine. 2018) it cannot allow us to attribute differences to a specific tissue compartment. Therefore, we believe our findings add value to the field of cortical development and give insight into the microstructural underpinnings of childhood development. As per the request of reviewer 1 (Q1), we have added more detail in the introduction on the interpretation of SANDI measures, and we direct the reviewer to this response to aid in the interpretation and utility of these novel measures in studying the developing brain.

Besides the aforementioned main comment I have, here are some suggestions which might improve the accessibility of this paper to general readership of Nature Communications:

1. The initial analyses of the dMRI and the later gene expression analyses in the result section are not in the same ROIs. This create a cognitive dissonance about what exactly regions they are referring to.

We appreciate that it may be difficult for the reader to interpret the findings of the two analyses together due to the different regions sampled. Since the spatial sampling of genetic data are limited and sparse, it is challenging to get direct correspondence of brain region location in each patient. However, to help with interpretability of the imaging results, we have now added a column to Table S1 which indicates the Yeo7network which corresponded to each of the brain regions sampled in the PsychENCODE dataset.

We have pointed out this change in the main text, section 4.2 (line 473), as follows:

“The cortical regions sampled are summarised in Table S1, alongside the approximate concordant Yeo7 parcel.”

2. The cell type specific trajectories are interesting but it puts a strong assumption on how gene expression changes over time in the bulk level. Better elaborate why this is a reasonable approach and empirically showcase the validity of such modeling.

Bulk tissue RNA-sequencing data is readily available and due to fewer time and technological constraints, generally allows larger sample sizes for analysis compared to single-cell methods. However, as the reviewer is alluding to, bulk tissue methods measure expression across heterogeneous tissue and may be confounded by changes in cell type or composition. We and others have previously used explored cell type expression profiles inferred from average expression of marker genes in bulk tissue RNA-seq (<https://www.nature.com/articles/s41467-020-17051-5>; <https://pubmed.ncbi.nlm.nih.gov/32276068/>; <https://pubmed.ncbi.nlm.nih.gov/28968835/>; <https://pubmed.ncbi.nlm.nih.gov/32450248/>). In the main manuscript we used bulk-tissue data due to the readily available nature of such databases, and the large number of samples available. Despite these efforts, we see the value in exploring the replicability of our findings in the context of single-cell RNA sequencing data.

To comprehensively address the reviewer’s comment about the validity of gene expression changes using bulk-tissue data, we took advantage of a recently published database (Velmeshev et al., 2023) of prenatal and postnatal human cortical development. This study explored >700,000 single-cell RNA sequencing profiles sourced from 169 tissue samples and 106 donors ranging from fetal (2nd trimester) up until adulthood. For every gene, they determined the cell-type, the trajectory of expression over development, and the period of maximal expression.

Using these data, which we accessed and downloaded for the purposes of this validation, we took a subset of these genes that had an age of onset in childhood (from age 4 years onwards), with maximal expression in late adolescence and/or adulthood. The resulting N=534 genes were predominately expressed in oligodendrocyte (Fig S12, label “OL”) cell-types (N=349) followed by excitatory neurons (Fig S12, label “L2_3”), which closely aligns with our bulk-tissue findings of an increase in oligodendrocyte and excitatory neuronal cell-type expression with age (Fig 3a,b of main manuscript). We believe this additional replication using a completely different dataset and technique agrees with our main conclusion that gene expression in oligodendrocyte cell-types increase over the period of childhood and adolescence.

Figure S12: Validation of our gene expression findings in an independent single-cell RNA dataset (Velmeshev et al., 2023). Cell-specific genes that had an onset of expression in childhood (>4 years) followed by a rapid increase into adolescence and adulthood were mainly expressed by oligodendrocytes.

We have described this additional validation in Section 2.2 of the results (line 182), as follows:

“To validate our bulk-tissue findings in an independent dataset, we took advantage of a recent single-cell RNA atlas of pre- and postnatal brain development (Velmeshev et al., 2023). Using these data, we identified a set of cell-specific genes with an onset of expression in childhood (>4 years) followed by a rapid increase through adolescence and into adulthood (n=534 genes). Most of these genes were expressed by oligodendrocytes (n=349; Figure S12), confirming our findings from bulk-tissue data.”

3. The simulations in the final section of the result is very interesting. But it is unclear how it related to the gene expression curves they aim to compare with.

We have added further information to the Results (section 2.3, line 237) to better describe these simulations in the context of the gene expression curves.

“To further evaluate the concordance between in vivo MRI and ex vivo gene expression patterns, we performed numerical simulations using realistic cell counts to explain the age-related patterns of the apparent soma radius. Assuming that the observed age-related slope of gene expression was proportional to the number of cells of each cell-type within an MRI voxel, we modelled cell-type composition changes based on the actual expected distribution of cell body radii within a voxel based on realistic cell counts and sizes. Our results revealed close correspondence between simulated and in vivo modelling results of R_{soma} (Fig S9), showing a 1% age-related decrease in both simulated and dMRI-derived data.”

4. The authors have added some clinical implications in the discuss section, but those are speculative and have no direct evidence coming from their analyses.

We agree that these clinical implications are not from direct evidence using the method we describe, so we have reframed this section as “Potential applications” to make it clearer that these applications could benefit from advanced cortical microstructural imaging. We have also modified the text in this section (line 351) to make this point clearer, as follows:

“Clinical implications Potential applications

Cortical morphology and myelination abnormalities are linked to various neuropsychiatric disorders (Chen et al., 2024) including schizophrenia (Alexander-Bloch et al., 2014; Wannan et al., 2019) which is characterised by deficiencies in myelination and oligodendrocyte production (Davis et al., 2003; Katsel et al., 2005). One potential future application is to quantify cortical microstructure in such clinical cohorts, especially with adaptations towards clinically feasible acquisition protocols (Barakovic et al., 2024; Margoni et al., 2023; Schiavi et al., 2023). Further strengthening this potential application, schizophrenia patients exhibit downregulation of myelination-related genes (Tkachev et al., 2003) and post-mortem studies have shown reduced oligodendrocyte density in layer 5 of dorsolateral prefrontal cortex compared to healthy controls (Kolomeets & Uranova, 2019). Additionally, young children with autism show age-related deficits in cortical T1w/T2w ratios (Chen et al., 2022). Future studies exploring these novel neuroimaging measures may provide valuable insights into cortical based abnormalities.”

Reviewer #3 (Remarks on code availability):

The URL contains multiple code repositories while the manuscript does not specify which one has been used. Therefore, it is impossible to review it.

Apologies, as this repository was not made public for review. We have now enclosed the code for the reviewer’s benefit. The github repository will be made public upon acceptance of the article.

References

- Alexander-Bloch, A. F., Reiss, P. T., Rapoport, J., McAdams, H., Giedd, J. N., Bullmore, E. T., & Gogtay, N. (2014). Abnormal cortical growth in schizophrenia targets normative modules of synchronized development. *Biol Psychiatry*, 76(6), 438-446.
- Barakovic, M., Weigel, M., Cagol, A., Schaedelin, S., Galbusera, R., Lu, P.-J., Chen, X., Melie-Garcia, L., Ocampo-Pineda, M., Bahn, E., Stadelmann, C., Palombo, M., Kappos, L., Kuhle, J., Magon, S., & Granziera, C. (2024). A novel imaging marker of cortical “cellularity” in multiple sclerosis patients. *Scientific Reports*, 14(1), 9848. <https://doi.org/10.1038/s41598-024-60497-6>
- Chen, B., Linke, A., Olson, L., Kohli, J., Kinnear, M., Sereno, M., Müller, R. A., Carper, R., & Fishman, I. (2022). Cortical myelination in toddlers and preschoolers with autism spectrum disorder. *Dev Neurobiol*, 82(3), 261-274. <https://doi.org/10.1002/dneu.22874>
- Chen, J., Patel, Z., Liu, S., Bock, N. A., Frey, B. N., & Suh, J. S. (2024). A systematic review of abnormalities in intracortical myelin across psychiatric illnesses. *Journal of Affective Disorders Reports*, 15, 100689. <https://doi.org/https://doi.org/10.1016/j.jadr.2023.100689>
- Davis, K. L., Stewart, D. G., Friedman, J. I., Buchsbaum, M., Harvey, P. D., Hof, P. R., Buxbaum, J., & Haroutunian, V. (2003). White Matter Changes in Schizophrenia: Evidence for Myelin-

Related Dysfunction. *Archives of General Psychiatry*, 60(5), 443-456.

<https://doi.org/10.1001/archpsyc.60.5.443>

- Fotiadis, P., Parkes, L., Davis, K. A., Satterthwaite, T. D., Shinohara, R. T., & Bassett, D. S. (2024). Structure–function coupling in macroscale human brain networks. *Nature reviews neuroscience*, 25(10), 688-704. <https://doi.org/10.1038/s41583-024-00846-6>
- Genc, S., Schiavi, S., Chamberland, M., Tax, C. M. W., Raven, E. P., Daducci, A., & Jones, D. K. (2024). Developmental differences in canonical cortical networks: insights from microstructure-informed tractography. *Network Neuroscience*, 1-48. https://doi.org/10.1162/netn_a_00378
- Genc, S., Tax, C. M. W., Raven, E. P., Chamberland, M., Parker, G. D., & Jones, D. K. (2020). Impact of b-value on estimates of apparent fibre density. *Human Brain Mapping*, 41(10), 2583-2595. <https://doi.org/10.1002/hbm.24964>
- Jones, D. K., Alexander, D. C., Bowtell, R., Cercignani, M., Dell'Acqua, F., McHugh, D. J., Miller, K. L., Palombo, M., Parker, G. J. M., Rudrapatna, U. S., & Tax, C. M. W. (2018). Microstructural imaging of the human brain with a 'super-scanner': 10 key advantages of ultra-strong gradients for diffusion MRI. *NeuroImage*, 182, 8-38. <https://doi.org/10.1016/j.neuroimage.2018.05.047>
- Katsel, P., Davis, K. L., & Haroutunian, V. (2005). Variations in myelin and oligodendrocyte-related gene expression across multiple brain regions in schizophrenia: A gene ontology study. *Schizophrenia Research*, 79(2), 157-173. <https://doi.org/https://doi.org/10.1016/j.schres.2005.06.007>
- Kolomeets, N. S., & Uranova, N. A. (2019). Reduced oligodendrocyte density in layer 5 of the prefrontal cortex in schizophrenia. *European Archives of Psychiatry and Clinical Neuroscience*, 269(4), 379-386. <https://doi.org/10.1007/s00406-018-0888-0>
- Lee, H., Lee, H.-H., Ma, Y., Eskandarian, L., Gaudet, K., Tian, Q., Krijnen, E. A., Russo, A. W., Salat, D. H., Klawiter, E. C., & Huang, S. Y. (2024). Age-related alterations in human cortical microstructure across the lifespan: Insights from high-gradient diffusion MRI. *Aging Cell*, 23(11), e14267. <https://doi.org/https://doi.org/10.1111/accel.14267>
- Margoni, M., Pagani, E., Preziosa, P., Palombo, M., Gueye, M., Azzimonti, M., Filippi, M., & Rocca, M. A. (2023). In vivo quantification of brain soma and neurite density abnormalities in multiple sclerosis. *Journal of Neurology*, 270(1), 433-445. <https://doi.org/10.1007/s00415-022-11386-3>
- Raven, E. P., Veraart, J., Kievit, R. A., Genc, S., Ward, I. L., Hall, J., Cunningham, A., Doherty, J., van den Bree, M. B. M., & Jones, D. K. (2023). In vivo evidence of microstructural hypo-connectivity of brain white matter in 22q11.2 deletion syndrome. *Molecular Psychiatry*. <https://doi.org/10.1038/s41380-023-02178-w>
- Schiavi, S., Palombo, M., Zacà, D., Tazza, F., Lapucci, C., Castellan, L., Costagli, M., & Inglese, M. (2023). Mapping tissue microstructure across the human brain on a clinical scanner with soma and neurite density image metrics. *Human Brain Mapping*, 44(13), 4792-4811. <https://doi.org/https://doi.org/10.1002/hbm.26416>
- Tkachev, D., Mimmack, M. L., Ryan, M. M., Wayland, M., Freeman, T., Jones, P. B., Starkey, M., Webster, M. J., Yolken, R. H., & Bahn, S. (2003). Oligodendrocyte dysfunction in schizophrenia and bipolar disorder. *Lancet*, 362(9386), 798-805. [https://doi.org/10.1016/s0140-6736\(03\)14289-4](https://doi.org/10.1016/s0140-6736(03)14289-4)
- Velmeshev, D., Perez, Y., Yan, Z., Valencia, J. E., Castaneda-Castellanos, D. R., Wang, L., Schirmer, L., Mayer, S., Wick, B., Wang, S., Nowakowski, T. J., Paredes, M., Huang, E. J., & Kriegstein, A. R. (2023). Single-cell analysis of prenatal and postnatal human cortical development. *Science*, 382(6667), eadf0834. <https://doi.org/10.1126/science.adf0834>
- Wannan, C. M., Cropley, V. L., Chakravarty, M. M., Bousman, C., Ganella, E. P., Bruggemann, J. M., Weickert, T. W., Weickert, C. S., Everall, I., & McGorry, P. (2019). Evidence for

network-based cortical thickness reductions in schizophrenia. *American Journal of Psychiatry*, 176(7), 552-563.

Yeo, B. T., Krienen, F. M., Sepulcre, J., Sabuncu, M. R., Lashkari, D., Hollinshead, M., Roffman, J. L., Smoller, J. W., Zöllei, L., Polimeni, J. R., Fischl, B., Liu, H., & Buckner, R. L. (2011). The organization of the human cerebral cortex estimated by intrinsic functional connectivity. *J Neurophysiol*, 106(3), 1125-1165. <https://doi.org/10.1152/jn.00338.2011>